

# Dynamics of North Balearic Front during an autumn Tramontane and Mistral storm: air-sea coupling processes and stratification budget diagnostic

Léo Seyfried[1], Claude Estournel[1], Patrick Marsaleix[1], and Evelyne Richard[1]

[1]Université de Toulouse, CNRS, UPS, Laboratoire d'Aérologie, Toulouse, France

*Correspondence to:* Léo Seyfried (leo.seyfried@aero.obs-mip.fr)

**Abstract.** The north Balearic front forms the southern branch of the cyclonic gyre in the North Western Mediterranean Sea. Its dynamics exhibits significant seasonal variability. During autumn, the front spreads northward during the calm wind periods and rapidly moves back southward when it is exposed to strong northerly wind events such as the Tramontane and Mistral. These strong winds considerably enhance the air-sea exchanges. To investigate the role of air-sea exchanges on the dynamics of the North Balearic front, we used observations and a high-resolution air-sea coupled modelling system, and focused on a strong wind event observed in late October 2012, which was well-documented during the Hydrological Cycle Mediterranean Experiment. The coupled model was able to correctly reproduce the $4°C$ sea surface temperature drop recorded in the frontal zone together with the observed southwestward displacement of the front. The comparison between the weak wind period preceding the event and the strong wind event itself highlighted the impact of the wind regime on the air-sea coupling, with both thermal and dynamical couplings during the low wind period and mainly thermal coupling during the strong wind period. The effect of air-sea exchanges on the stratification variations in the frontal zone was investigated with a stratification budget diagnosis. The stratification variations are controlled by diabatic air-sea buoyancy flux, adiabatic Ekman buoyancy flux, and advective processes. During the strong wind period, the Ekman buoyancy flux was found to be three times greater than the air-sea buoyancy flux and thus played a major role in the destratification of the frontal zone. The role of Ekman pumping and inertial wave on the advective processes is also discussed.



# 1 Introduction

The North Balearic Front (NBF) is an extension of the Balearic Current (BC), from the Balearic Sea to the Ligurian Sea (Font et al., 1988). The NBF forms the south branch of the surface oceanic cyclonic gyre of the North Western Mediterranean Sea (NWMS), closed to the north and west by the Northern Current (Millot, 1999), and to the east by the West Corsica Current (WCC) (Fig. 1). This surface density front (100-200 m deep) separates the warm and fresh Atlantic Water (AW) which has recently entered the south of the basin from the colder and saltier AW present in the center of cyclonic gyre (Millot and Taupier-Letage, 2005). This front forms a "Lagrangian barrier" (Mancho et al., 2008) which plays an important role on the nutrients budget and planktonic ecosystem (Estrada et al., 1999) and marine ecosystems distributions (Gannier and Praca, 2007; Cotté et al., 2011).

The NBF dynamics is strongly influenced by mesoscale and submesoscale structures with significant seasonal variations (García et al., 1994). During winter the surface front is located to the south (around 40 °N) because of the densification of surface waters by open-ocean deep convection that occurs in the center of the gyre (Marshall and Schott, 1999). During summer the light AW present in the south of the basin spreads northward and the position of the front is further north. The short time scales variations of the surface front can also be significant in response to the wind variability: when the northern wind is weak, light water spreads northward over the dense water while when the northern wind is strong the front shifts southward (Estournel et al., 2016a).

This study focuses on the NBF dynamics during an autumn storm. This storm occurs during the Intensive Observation Period (IOP) number 16 (26-29 October 2012) of the HyMeX program (Drobinski et al., 2013; Ducrocq et al., 2013; Estournel et al., 2016b). This IOP can be divided into two parts, the first part IOP16a (26 October 2012) was dedicated to heavy precipitation events whereas the second part IOP16b (27-29 October 2012) focussed on strong wind events. During IOP16a the convergence between southwesterly and southeasterly flows initiated and maintained strong precipitation over the southeastern French coasts (Duffourg et al., 2016). During IOP16b a severe northerly wind event, associated with Tramontane and Mistral, and a large decrease of the Sea Surface Temperature (SST) were observed at the Lion meteorological buoy (Lebeaupin Brossier et al., 2014). Lebeaupin Brossier et al. (2014) explained this large SST drop by the intense surface heat losses producing cooling of surface water and vertical mixing. Estournel et al. (2016a) showed that the large decrease of SST at the buoy was also associated with the rapid southward displacement of the NBF.

The Tramontane and Mistral are two strong, dry and cold north-westerly/northerly winds which are channeled by the topography, between the Pyrenees and the Massif Central for the Tramontane, and between the Massif Central and the Alps for the Mistral (Fig. 1). These winds produce strong air-sea exchanges (Flamant, 2003; Hauser et al., 2003) which lead to important diabatic buoyancy losses for the sea and which also impact the NBF position in autumn (Seyfried et al., 2017). Furthermore, the Tramontane and Mistral have up- or down-front winds components over the NBF and the Northern Current. Down (up) front wind corresponds to a wind parallel and in the same (opposite) direction as the current (Fig. 1). In an academic study, Thomas and Ferrari (2008) showed that the up or down front wind can respectively restratify or destratify the mixed layer. This modification of stratification is due to the differential advection of density by the Ekman transport which destabilizes the front





and releases the symmetric instability (Taylor and Ferrari, 2010). To represent the modification of stratification generated by the Ekman transport, Thomas and Lee (2005) defined a surface adiabatic buoyancy flux named the Ekman buoyancy flux as the product between the Ekman transport and the horizontal surface buoyancy gradient. This physical diagnostic was applied to the real case of Kuroshio (Rainville et al., 2007) and Gulf Stream (Thomas et al., 2016), showing the major role of the Ekman

buoyancy flux as compared to the diabatic buoyancy flux on the modification of stratification in frontal zones. In the NWMS, a realistic modelling study (Giordani et al., 2016) also analysed the respective roles of the diabatic air/sea and adiabatic Ekman buoyancy losses along the Northern Current front on the deep water formation during winter. This study found that the Ekman buoyancy flux contribution was dominant along the northern branch of the Northern Current.

In the NWMS (Lebeaupin Brossier and Drobinski, 2009; Small et al., 2012; Renault et al., 2012; Rainaud et al., 2017)

showed the importance of using air-sea coupled modelling to represent the rapid variations of air-sea exchanges induced by the decrease of SST during Tramontane and Mistral events. Furthermore, in frontal zones, the air-sea exchanges are clearly an air-sea coupled process. The SST front and sea surface frontal currents directly impact the air-sea exchanges with significant feedback between the sea and the atmosphere (Small et al., 2008). The atmospheric response to the front creates wind stress divergence and curl impacting the sea by Ekman pumping (Chelton and Xie, 2010). These air-sea feedbacks in frontal zones

may also have a marked influence on the diabatic and Ekman buoyancy fluxes (Thomas and Lee, 2005).

The first aim of this paper is to describe the dynamics of the NBF during IOP16 of the HyMeX programme. Our approach combines satellite data, in situ data and air-sea coupled simulation at kilometric scale. A second aim is to evaluate the air-sea exchanges in the frontal zone and investigate their impact on ocean stratification. In particular, the competing roles of the diabatic buoyancy flux, the adiabatic Ekman buoyancy flux, and advective processes will be assessed by means of an original

stratification budget diagnosis.

This paper is organized as follows. Section 2 presents the IOP16 case study. In Section 3, the results of the air-sea coupled simulation are analysed and discussed with respect to the available observations. Then, the NBF dynamics and associated air-sea exchanges are investigated in Sections 4 and 5, respectively. The computation and results of the stratification budget diagnosis are presented in Section 6. Finally, the results are discussed and some conclusions are drawn in Section 7.

## 25  2   Case study

### 2.1   Atmospheric conditions

Figure 2 shows the Mean Sea Level Pressure (MSLP) and the surface air temperature and wind from the European Centre for Medium-range Weather Forecasts (ECMWF) analysis at 1200 UTC for IOP16a (26 October 2012) and IOP16b (27, 28 and 29 October 2012). IOP16a (Fig. 2.a) was characterized by low pressure centred over the Pyrenees and a southwesterly wind

advecting warm air (temperature above 20 °C) over the NWMS. On the next day (Fig. 2.b), the low pressure was positioned over the Alps, leading to flow reversal with northwesterly/northerly winds advecting cold air over the NMWS. On 28 October (Fig. 2.c) the surface low was deeper and located over the Ligurian Sea. The northwesterly/northerly wind increased and





extended over the western Mediterranean basin with an air temperature lower than 10 °C in the NWMS. Finally, on 29 October (Fig. 2.d) the cyclone dissipated and the air temperature increased again.

The Lion meteorological buoy, positioned in the abyssal plain of the Gulf of Lion (4.64°E - 42.06°N, indicated by the black triangle in Fig. 2) and in the Tramontane and Mistral flows, provides hourly measurements of the 10-m wind, the 2-m temperature and humidity, and the SST (Fig. 3). During IOP16a (26 October 2012) the Lion buoy measured low wind speeds between $5\,\mathrm{m\,s^{-1}}$ and $10\,\mathrm{m\,s^{-1}}$. The 2-m temperature and relative humidity remained nearly constant around 20°C and 80%, respectively. At 0900 UTC a brief decrease in wind and 2-m temperature was observed, corresponding the signature of a precipitation event. Then, on 27 October, the wind speed increased from $8\,\mathrm{m\,s^{-1}}$ at 0600 UTC to a maximum of $26.2\,\mathrm{m\,s^{-1}}$ at 1800 UTC. On 28 October, the wind speed remained nearly constant around $22\,\mathrm{m\,s^{-1}}$ and finally decreased on 29 October to less than $5\,\mathrm{m\,s^{-1}}$ at the end of the day. A rapid decrease of the 2-m temperature was observed on 27 October during the wind increase, with a temperature drop of 7.5°C in a few hours. After this rapid decrease, the 2-m temperature continued to fall slowly, reaching a minimum of 8°C on 28 October at 0900 UTC before rising slowly by 3°C on 29 October. During IOP16b, the 2-m humidity decreased from 80% at 0000 UTC on 27 October to 40% at 0000 UTC on 30 October.

To summarize, this autumn Tramontane and Mistral storm (27, 28 and 29 October 2012) was characterised by northerly / northwesterly winds exceeding $20\,\mathrm{m\,s^{-1}}$. Meanwhile, the surface air temperature and humidity dropped by 10 °C and 40%, respectively.

## 2.2 Oceanic conditions

Figure 4 (a-f) shows two SST analysis products from satellite observations before and after the strong wind event, and the SST difference. The first analysis is the global OSTIA product (Donlon et al., 2012), the horizontal resolution of which is about 6 km. This product is used in the ECMWF operational model. The second analysis is the Mediterranean Copernicus product (Buongiorno Nardelli et al., 2013) at higher resolution (about 1 km). There is no significant difference between these two data sets. The NBF extends over several tens of kilometres in both products with steeper gradients in the high resolution product. At 0000 UTC on 25 October (Fig. 4 a and d) the NBF separated the northern water at 18.5 °C from the southern water at 21.5 °C, yielding a surface thermal front of 3 °C. The Lion buoy was positioned in the warm side of the frontal zone. At the end of IOP16 (30 October 2012, Fig. 4 b and e) the NBF had moved several tens of kilometres to the south and the Lion buoy was then positioned in the cold side of the frontal zone. During IOP16, the SST decrease (Fig. 4 c and f) was about 2°C in the basin and reached more than 3°C around the Lion buoy. The temperature of the southern water (northern water) was around 16°C (20°C), establishing a surface thermal front of 4°C. At the Lion buoy, the SST in-situ measurement (Fig. 3.d) showed a rapid decrease in SST of about 4°C between 1200 UTC on 27 October and 0000 UTC on 29 October in agreement with the analyses.

Gliders provided information on the oceanic stratification and the spatial distribution of water masses during IOP16 (see their position on Fig. 4). Two gliders were present near the Lion buoy on 25 October 2012, in the frontal zone (Fig. 5). During the IOP, the first glider, named Eudoxus, moved to the west (section A-B in Fig. 5). The second glider, named Campe, moved to the south (section C-D in Fig. 5). Before the strong wind, according to the two gliders, the Mixed Layer Depth (MLD) near the Lion buoy was about 40 m, the potential temperature was higher than 19.5°C in the mixed layer (Fig. 5 d and f) and the





salinity was lower than 38.2 (not shown). This temperature corresponds to the surface temperature of the cold side of the NBF (Fig. 4 a and d). According to the Eudoxus glider (Fig. 5.d), between the two ends of the section, the difference in the MLD was about 60 m and the temperature drop about 3.5°C. These differences resulted from both the crossing of the NBF and the onset of the strong wind. According to the Campe glider (Fig. 5.f), which remained on the southern side of the front, the difference

5    in the MLD was 40 m and the temperature increased by 0.5°C. However, along this glider track, the 2°C temperature drop measured between October 27 and 29 was probably due to the strong wind event whereas the temperature increase observed after 29 October resulted from the southward displacement of the glider (see Fig. 4.e).

Furthermore, two Argo profiles (see their position on Fig. 4) were available during IOP16 in the frontal zone (Fig. 6). The first profile was obtained on 26 October, 40 km east of the Lion buoy (41.8°N-5.05°E) and the second one on 31 October, 30

10    km east of the Lion buoy (41.8°N-4.86°E). In the mixed layer, the potential temperature drop between the two profiles was large (about 5°C) while the salinity decrease was about 0.05. Finally, the potential density increase (about 1.32 $\mathrm{kg\ m^{-3}}$ at the surface) was driven directly by the potential temperature decrease and the MLD deepened by only a few metres.

In summary, the observations showed a rapid decrease of the SST (greater than 4°C) in the frontal zone. This rapid decrease was associated with the southward displacement of the NBF and a slight deepening of the MLD in the frontal zone. As satellite

and in situ data give only a partial view of the NBF dynamics, and in order to better describe and understand this process, a high resolution air-sea coupled simulation was performed.

## 3    Numerical experiment

### 3.1    Air-sea coupled simulation

To perform the simulation, we used the Meso-NH - SURFEX - Symphonie system (Meso-NH: Lafore et al. (1997), SURFEX:

Masson et al. (2012), Symphonie: Marsaleix et al. (2008, 2012); Michaud et al. (2012)) based on the SURFEX-OASIS interface (Voldoire et al., 2017). This coupled system was described and validated in Seyfried et al. (2017).

A high-resolution configuration of this coupled system was implemented with a horizontal resolution of 1 km in the oceanic model and of 2.5 km in the atmospheric model. At this resolution, the oceanic model can be considered as eddy-resolving. In the NWMS, the Rossby radius is of the order of 5 to 10 km (Beuvier et al., 2012), which corresponds to the effective resolution

of the oceanic model. Bosse (2015) showed the ability of the Symphonie oceanic model to reproduce the symmetric instability process at this resolution. However, at this resolution, the oceanic model convection must be parameterized, as described in Estournel et al. (2016a). The non-solar heat fluxes and water fluxes are actually re-distributed over the whole mixed layer, in accordance with Deardorff et al. (1969). The atmospheric model at 2.5 km can be considered as a convection-permitting model i.e. the convection is directly resolved by the atmospheric model equations. In the vertical, the oceanic model uses 40

generalized sigma vertical levels, 10 of them in the first hundred metres (above the abyssal plain) with a resolution just below the sea surface of 1.5 metres. The atmospheric model uses 52 terrain-following vertical levels ranging from 15 to 15000 m.

Using the SURFEX OASIS3-MCT interface (Voldoire et al., 2017), the SST and sea surface current are sent to the SURFEX model, which then returns the wind stress ($\tau$), the shortwave flux ($SW$), the non-solar heat fluxes ($Q_{ns} = LW + H + L$



with $LW$ the longwave flux, $H$ the sensible heat flux and $L$ the latent heat flux) and the water fluxes ($E - P$, Evaporation minus Precipitation). The turbulent fluxes are calculated using the bulk parameterization developed by Moon et al. (2007). The coupling frequency is set to 10 minutes. Flux calculations are carried out on the atmospheric model grid (after bilinear interpolation of the oceanic fields) and are then bilinearly interpolated on the grid of the ocean model.

The atmospheric model covers the whole western Mediterranean basin (same spatial coverage as in Seyfried et al. (2017)). The ocean model covers the western Mediterranean Sea, excluding the Alboran Sea and part of the Tyrrhenian Sea (same grid as in Seyfried et al. (2017)). In the oceanic part of the atmospheric grid not covered by the ocean grid, the air-sea fluxes are computed using the sea surface temperature provided by the SST OSTIA product (Donlon et al., 2012) and without using sea surface current.

The coupled simulation started at 0000 UTC on 25 October 2012 (24 hours before the beginning of IOP16) and ended at 0000 UTC on 30 October 2012 (at the end of IOP16). The ocean model was initialized with a coupled simulation as described in Seyfried et al. (2017) (of all the simulations presented in this paper, the so-called "MOON" simulation was chosen for its better agreement with the available observations). Spin up was avoided as the initial state given by the "MOON" simulation was based on the same oceanic model configuration and grid. The boundary conditions of the ocean model were provided by

the analyses of the MERCATOR-OCEAN operational centre (Lellouche et al., 2013) based on the NEMO model (Maraldi et al., 2013). Initial and boundary conditions of the atmospheric model were provided by the 6-hour ECMWF analyses with a horizontal resolution of 1/8°.

## 3.2   Simulation validation

The simulation was validated with in situ and satellite observations. At the Lion buoy (Fig. 3.a, b and c), the simulation correctly

reproduced the wind intensity evolution but the wind absolute maximum at 1800 UTC on 27 October was underestimated by 5 m s$^{-1}$. The 2 m temperature and humidity were also well represented, but the rapid decrease in temperature observed around the wind maximum was slower in the model and a positive humidity bias of 10 % appeared at the end of the simulation.

    Then, to validate the SST decrease in the NWMS and the NBF dynamics, the SST satellite analyses and the simulated SST were compared (Fig. 4). The simulated temperature field contained more small-scale structures (fronts, eddies, filaments) than

the corresponding satellite field. The horizontal resolution of the latter was, in fact, substantially lower than that of the model. The initial surface temperature and NBF position were well represented by the simulation but the extension of the warm AW to the east of the Lion buoy was greater than in the observations. At the end of IOP16, the SST decrease and the NBF southward shifting were also well represented even if the latter was less pronounced than in the observations. At the Lion buoy (Fig. 3.d) the SST decrease of 4°C was well represented by the simulation but the decrease started 6 hours earlier.

Finally, to validate the evolution of oceanic stratification, the simulation results were compared to the profiles obtained with the gliders and Argo (Figs 5 and 6). The comparison with the gliders showed that the initial state of the model was satisfactory (Fig. 5) near the Lion buoy even though the MLD was less deep (by about 20 m) and the thermocline thicker, probably because of the insufficient vertical resolution of the model. In comparison with results from the first glider (Eudoxus), which moved westward, the SST and MLD deepening were well represented. However, at the extremity of the section, the MLD



was deeper than in the observation, probably because of the crossing of the Northern Current visible from the slope of the isopycnals, which differed in the simulation and the observations. For the second glider (Campe), which moved southward, the temperature evolution was represented differently by the glider and the simulation due to the presence of small-scale structures in the simulation that were not observed. During strong wind events, the temperature decrease was less marked in the simulation

but the MLD deepening was well represented. Finally, after the strong wind event (29 October), the temperature increase was not simulated. The Argo comparison (Fig. 6) showed good agreement for the initial temperature profile, while a surface negative salinity bias of about 0.2 was shown by the model. At the end of the strong wind event, the simulation showed a decrease of the mixed layer temperature and an increase in salinity and density that were smaller than in the observations, except for salinity. However, the observed profile was obtained to the north of the NBF whereas, in the simulation, it corresponded to the frontal

zone (see their position on Fig. 4). The temperature, salinity and density simulated to the north of the NBF (41.8°N -5.15 °E, green dashed line in Fig. 6) were much closer to the Argo observations.

To conclude, this simulation is satisfactory regarding the atmospheric and oceanic evolution, including air and sea parameters at the surface, the oceanic mixed layer deepening, and the NBF southward shift during the strong wind event. In addition, the 1 km resolution oceanic simulation correctly represents the narrow surface front, and associated meso and submesoscale

structures, the position of which obviously differ from the actual ones.

## 4    Characteristics and dynamics of NBF

The NBF dynamics during IOP16 (26 October 2012 to 30 October 2012) is now examined through an analysis of the air-sea coupled simulation.

Figure 7 (a-b) represents the surface potential density spatial distribution before (0000 UTC on 26 October) and after (0000

UTC on 30 October) the IOP16 and Fig. 7 (c) shows the density difference between these two dates. Two specific surface water masses of the NWMS (the relatively light AW to the south and the dense AW to the north) clearly appear on Fig. 7 a and b (in blue and red respectively). Before IOP16 (Fig. 7.a), the Atlantic waters protrude northward forming a meander of light water extending to 42°N and from 3°E to 6°E. The difference of surface density between the two sides of the front is about 1 $\mathrm{kg\,m^{-3}}$ with a surface density gradient locally greater than 0.1 $\mathrm{kg\,m^{-3}}$ per kilometre. Filaments formed at the periphery of

the meander extend northward. Further to the east (around 6.5°E) a second intrusion of light water occurs, but its northward extension is smaller and its density front weaker. These two meanders shape the NBF, which can be characterized by the 27.0 $\mathrm{kg\,m^{-3}}$ surface isopycnal. After IOP16 (Fig. 7.b), the light water patch is shifted to the southwest and the surface potential density increases in the NWMS. The NBF surface density gradient is less strong, except along the western part of the meander, and filament structures have dissipated. The NBF is now characterized by the 28.0 $\mathrm{kg\,m^{-3}}$ surface isopycnal. Looking at the

surface density evolution between the beginning and the end of IOP16, the maximum potential density increase (Fig. 7.c), larger than 1.5 $\mathrm{kg\,m^{-3}}$, is located in the area impacted by the NBF dynamics (along the meander) and along the north coast of the Gulf of Lion. Note the appearance of cold, dense waters along the French coast (around 5.5°E), a sign of coastal upwelling.





The cumulative effect of the strong wind event on the stratification of the upper layers can be analysed through the Stratification Index (SI) (Estournel et al., 2016a).

$$SI(H) = \int\limits_{H}^{0} (\rho(H) - \rho(h))dh \tag{1}$$

*where $\rho$ is the potential density* (kg m$^{-3}$) *and $H$ the reference level* (m)*. SI is expressed in* kg m$^{-2}$. *It represents the amount*

*of buoyancy to be extracted to mix the water column from the surface to level $H$ and achieve a homogeneous density $\rho(H)$.*

Figure 7 (d-f) shows the SI at 250 m (depth below which no significant changes are observed), before and after IOP16, and the SI differences between these two periods. Before the IOP16 (Fig. 7.d), the light Atlantic Water corresponds to higher stratification, with an SI at 250 m greater than 120 kg m$^{-2}$. Inside this meander, a mesoscale anticyclonic eddy has a notable effect on stratification with an SI higher than 200 kg m$^{-2}$ (around 41°N and 4.5°E). The NBF separates this patch of highly

stratified water from the less stratified Modified Atlantic Water characterized by an SI lower than 90 kg m$^{-2}$, i.e. a stratification front of about 30 kg m$^{-2}$. The filaments visible in the surface potential density map (Fig. 7.a) are too thin to have a significant impact on the upper layer integrated stratification. The SI evolution during IOP16 (Fig. 7.f) shows a maximum loss of stratification (about 50 kg m$^{-2}$) in the northern and eastern parts of the NBF meander and a stratification gain (about 20 kg m$^{-2}$) in the western part of the NBF meander. Outside the frontal region, SI variations appear to be related to the displacement of

mesoscale and submesoscale eddies.

Two vertical sections of potential density are now examined to illustrate the vertical processes. Figure 8 presents the vertical distribution of isopycnals before (black lines in Fig. 8) and after (blue lines in Fig. 8) IOP16 and the sea water potential density difference (in colours) along North-South (NS) and East-West (EW) sections (indicated by black lines in Fig. 7).

On the NS section (Fig. 8.a), before the strong wind event, the section intersects the NBF meander at 42.1°N (the NBF

position is defined by the 27.0 kg m$^{-3}$ isopycnal). South of the NBF, a patch of light AW, with a density lower than 27.0 kg m$^{-3}$, extends over a depth of about 50 m. To the north, the light water thins to 10 m around 42.4°N, corresponding to the filament structure. This section also intersects the mesoscale anticyclonic eddy previously described around 41°N (explaining the local deepening of the isopycnals), and the Northern Current along the bathymetric slope. Between 41.75°N and 42.75°N, the doming of the isopycnals corresponds to the cyclonic gyre. After the strong wind event, the NBF migrates southward by

about 0.3° (about 35 km) and the NBF position is now defined by the 28.0 kg m$^{-3}$ isopycnal. The maximum increase of potential density (more than 1.5 kg m$^{-3}$ to a depth of 40 m) is obtained in the displacement area of the NBF.

The EW section (Fig. 8.b) before IOP16, which crosses the meander (Fig. 7c), intersects the NBF twice, at about 4.0°E and 6.0°E. The isopycnal doming, corresponding to the cyclonic gyre, is also present between 3.4°E and 4.4°E, while the dive of the isopycnals to the west indicates the change of direction of the Northern Current, which follows the isobaths of the

30 continental slope. After IOP16, the eastern intercept point of the NBF with the section of Fig. 8.b has been shifted westward by about 1.0° (about 80 km). The western intercept point has also moved westward but only by about 0.2° (about 16 km). The maximum increase of potential density is obtained in the eastern part of the NBF, with an increase greater than 1.0 kg m$^{-3}$ to



a depth of 30 m. The density increase in the heart of the Atlantic water (between 3.8°E and 5.2°E) is smaller ($< 1$ kg m$^{-3}$ at 40 m). Near the western part of the NBF (around 3.7°E), the density does not increase at the surface and decreases at 50 m depth due to the isopycnal deepening, which explains the SI increase in this region (Fig. 7.f).

To conclude, during the strong wind event, a horizontal displacement of the NBF of several tens of kilometres is simulated. This directly impacts the evolution of oceanic stratification in the NBF zone. The stratification evolution is also impacted by vertical advective processes producing isopycnal deepening.

## 5 Air-sea exchanges

The air-sea exchanges can be divided into three fluxes: the wind stress ($\tau$, in N m$^{-2}$), the net heat flux ($Q_{net} = SW + LW + H + LE$, in W m$^{-2}$) and the water flux ($E - P$, in mm h$^{-1}$).

Figure 9 shows the time evolution of wind stress, net heat flux and water flux calculated at the Lion buoy, positioned in the NBF zone. The wind stress (Fig. 9.a) is directly correlated to the wind speed (Fig. 3.a), with a maximum wind stress higher than 1.5 N m$^{-2}$. On average, the net heat flux (Fig. 9.b) was close to zero between 25 and 26 October. During this low wind period the solar heat flux balanced the non-solar heat flux. During the strong wind event, upward (i.e. cooling the sea) net heat fluxes were correlated with the wind speed, with a maximum net heat flux close to 2000 W m$^{-2}$ on 28 October 2012. The water flux (Fig. 9.c) was impacted by the high precipitation event occurring on 26 October, with a maximum downward instantaneous water flux of about 25 mm h$^{-1}$. During the strong wind event, evaporation led to an increase in density.

The spatial distribution of air-sea fluxes is now examined. Figure 10 represents the wind stress, net heat flux and water flux averaged over IOP16a (26 October 2012) and IOP16b (27-29 October 2012). During IOP16a the NWMS is dominated by weak southwesterly winds (Fig. 10.a). During this period, the NBF directly impacts the wind stress distribution. The maximum of wind stress appears to the south of the front over the patch of warm AW. The net heat fluxes (Fig. 10.c) are also directly impacted by the NBF position. The maximum heat loss appears on the warm patch, due to the higher SST, directly impacting turbulent heat fluxes. Downward water fluxes (Fig. 10.e) are organized in bands during this high precipitation event. During IOP16b, the NWMS is dominated by strong northerly/northwesterly winds (Fig. 10.b), corresponding to Tramontane and Mistral events. The maximum wind stress is located in these wind veins, off the Gulf of Lions. The NBF has no clear impact on the wind stress spatial distribution. On the other hand, the net heat flux and water flux are directly impacted by the NBF position, with a maximum heat and water loss higher than 1200 W m$^{-2}$ and 30 mm day$^{-1}$ on the warm side of the front, and cross frontal differences greater than 200 W m$^{-2}$ and 10 mm day$^{-1}$.

The wind stress curl during IOP16a and IOP16b are presented on Fig. 11. As explained by Chelton and Xie (2010), surface winds are weaker on the cold side of a thermal front. This produces a divergence when the wind blows across the front and a curl when the wind blows parallel to the front. In the case of the NBF meander during the low wind period of IOP16a, Fig. 11.a indicates the wind curl modifications linked to the front. In the eastern part of the meander, a negative along-front wind stress curl appears clearly and, in the western part of the meander, a positive along-front wind stress curl also appears clearly. During IOP16b the wind stress curl appears principally around the Tramontane and Mistral wind corridors. A positive (negative) wind





stress curl corresponds to the left cyclonic (right anticyclonic) side of the wind vein. In the eastern part of the NBF meander (around 5°E and 41.8°N) a positive wind stress curl appears. However, it is difficult to know whether the wind stress curl is directly connected to the NBF meander or to the fine wind jet structures. Finally, during the strong wind event, unlike the situation in the low wind event, the NBF meander does not appear to have a major influence on the wind stress curl.

To conclude, the NBF position and dynamics directly impact the spatial and temporal distributions of air-sea heat and water exchanges. Furthermore, during the low wind period, the NBF also impacts the wind stress with the generation of along-front wind stress curl.

## 6   Stratification budget diagnosis

As shown by Thomas and Lee (2005), down-front wind can destabilize the water column by cross-front advection of density

by Ekman flows. Following Thomas and Lee (2005) the friction induced buoyancy flux or Ekman Buoyancy Flux (EBF) is given by:

$$EBF = -\frac{g}{\rho_0} \boldsymbol{M_e} . \boldsymbol{\nabla_h} \rho_{(z=0)} \tag{2}$$

*where EBF is the Ekman Buoyancy Flux in* $\mathrm{m^2\,s^{-3}}$, $\boldsymbol{M_e}$ *the Ekman transport (Eq. 3) in* $\mathrm{m^2\,s^{-1}}$ , $\boldsymbol{\nabla_h}$ *the horizontal gradient and* $\rho_{(z=0)}$ *the surface density in* $\mathrm{kg\,m^{-3}}$.

$$\boldsymbol{M_e} = \frac{\hat{z} \times \boldsymbol{\tau}}{\rho_0 \zeta_a} \tag{3}$$

*where* $z$ *is the vertical unit vector,* $\boldsymbol{\tau}$ *the wind stress in* $\mathrm{N\,m^{-2}}$, *and* $\zeta_a$ *the absolute vorticity in* $\mathrm{s^{-1}}$.

It is interesting to compare this flux to the buoyancy flux induced by diabatic processes and given by (Mertens and Schott, 1998):

$$B_0 = g\alpha \frac{Q_{net}}{\rho_0 C_p} + g\beta SSS F_w \tag{4}$$

*where* $B_0$ *is the diabatic buoyancy flux in* $\mathrm{m^2\,s^{-3}}$, $\alpha$ *the thermal expansion coefficient in* $\mathrm{K^{-1}}$, $C_p$ *the specific heat capacity in* $\mathrm{J\,kg^{-1}\,K^{-1}}$, $\beta$ *the saline contraction coefficient, SSS the sea surface salinity, E the evaporation and P the precipitation in* $\mathrm{m\,s^{-1}}$.

In order to evaluate the competing roles of the diabatic buoyancy flux and the Ekman buoyancy fluxes on stratification evolution we assume that the stratification index at depth H between the times T1 and T2 can be written as :

$$\Delta SI(H) = SI(H)_{T2} - SI(H)_{T1} = \frac{\rho_0}{g} \int_{T1}^{T2} B_0(t).dt + \frac{\rho_0}{g} \int_{T1}^{T2} EBF(t).dt + R \tag{5}$$





*where, the first and second right hand side terms represent the mass fluxes, induced by diabatic and friction processes respectively, whereas the third term induce all the remaining processes.*

Figures 12 and 13 present the spatial distribution of the different terms of the stratification budget diagnosis (Eq. 5) for IOP16a and IOP16b. During the low wind period (IOP16a), the stratification (Fig. 12.a) decreases (increases) along the western (eastern) part of the NBF meander, by about 20 kg m$^{-2}$. This stratification evolution is not directly controlled by the diabatic and Ekman buoyancy mass fluxes (Fig. 12 b and c), which are small relative to the residual term (Fig. 12.d). The advective processes play a dominant role in the evolution of stratification during this period of low wind. During the strong wind event (IOP16b), in contrast to the low wind period, the stratification (Fig. 13 a) decreases (increases) along the eastern (western) part of the NBF meander, by about 60 kg m$^{-2}$ (20 kg m$^{-2}$). The evolution of the stratification (Fig. 13.a) is directly controlled by the diabatic and Ekman buoyancy mass fluxes (Fig. 13 b and c). The cumulated diabatic buoyancy mass flux loss (Fig. 13.b) is between 15 and 30 kg m$^{-2}$ in the Tramontane and Mistral corridor, being slightly larger south of the NBF (not shown on Fig. 13.b but visible for the net heat flux on Fig. 10.c). However, stratification evolution in the NBF meander zone is principally driven by the Ekman buoyancy mass flux (Fig. 13.c). In the east (west) of the NBF meander, a down-front (up-front) wind generates an Ekman buoyancy flux of -30 to -45 kg m$^{-2}$ (15 to 30 kg m$^{-2}$). In a second order, the residual term (Fig. 13.d) also plays a role in how stratification evolves in the NBF zone, particularly in the western part of the NBF meander, with a stratification gain.

Figure 14 shows the time evolution of the different terms of the stratification budget (Eq. 5) in the eastern and western parts of the NBF meander. Firstly, the presence of oscillations produced by inertial waves can be noted on the evolution of the stratification. These oscillations have a period of about 17 hours and amplitude of about 10 kg m$^{-2}$. Except for these oscillations, the stratification remains approximately constant from 25 to 27 October (IOP16a). During IOP16b, the stratification decreases (increases) along the eastern (western) part of the NBF meander, from 122 kg m$^{-2}$ to 69 kg m$^{-2}$ (from 86 kg m$^{-2}$ to 102 kg m$^{-2}$). Along the east of the NBF meander (Fig. 14.a), the stratification decrease (53 kg m$^{-2}$) is due to the buoyancy fluxes and, more precisely, to the Ekman buoyancy flux, (45 kg m$^{-2}$ compared to 17 kg m$^{-2}$ for diabatic buoyancy mass flux). In this region, the advective processes remain negligible (about 1 kg m$^{-2}$). Along the west of the NBF meander (Fig. 14.b), the stratification decreases by about 5 kg m$^{-2}$ during the low wind period and increases by about 20 kg m$^{-2}$ during the strong wind period. The stratification decrease during the low wind period is driven by a negative Ekman buoyancy flux. During the strong wind period, the Ekman buoyancy flux of 20 kg m$^{-2}$ balances the diabatic buoyancy loss of 18 kg m$^{-2}$. The residual term of about 18 kg m$^{-2}$ plays a major role on the stratification increase in this zone.

The residual term is not negligible in our stratification budget diagnosis, particularly along the western part of the NBF meander. A source of this term is the vertical advection generated by the Ekman pumping directly connected to the wind stress curl (Eq. 6).

$$W_e = \frac{\nabla \times \boldsymbol{\tau}}{\rho_0 \zeta_a} \qquad (6)$$



To illustrate this, Fig. 15 shows the $28.5 \ \mathrm{kg \ m^{-3}}$ isopycnal depth variation and the Ekman pumping averaged over IOP16a and IOP16b. During IOP16a, the isopycnal depth variation (Fig. 15.a) is directly anticorrelated with the residual term of the stratification budget (Fig. 12.d). The stratification decreases in ascending isopycnal zones and increases in subsiding isopycnal zones. This anticorrelation illustrates the major role of the vertical processes in the residual term. The Ekman pumping, gener-

ated by the interaction between the wind and the front, is an important source of this vertical advection (Fig. 15.c). Although, at the scale of the whole domain, the Ekman pumping does not appear to be correlated with the change in the level of the isopycnals, this is not the case along some parts of the meander such as its western part, where the angle between the wind and the isopycnals is small and the Ekman pumping seems to contribute to the change of stratification. During IOP16b, the isopycnal depth variations (Fig. 15.b) are also anticorrelated with the residual term of the stratification budget (Fig. 13.d). In the western,

unlike in the eastern, part of the NBF meander, the isopycnal deepening is clearly produced by the Ekman pumping (Fig. 15.d). Another source of vertical advection in frontal zones is the ageostrophic secondary circulation cell generated by down-front wind and frontogenesis processes (Thomas and Lee, 2005), with upward and downward vertical currents respectively in the light and dense parts of front. It is probably this circulation that impacts the isopycnal depth in this zone.

In conclusion, to a first-order approximation, the stratification variations in the NBF zone during IOP16 are principally

driven by the Ekman buoyancy flux generated by the interactions between the strong wind event and the NBF. However, the residual term of stratification budget diagnosis is not negligible, particularly in the western part of the front through an Ekman pumping effect.

## 7   Summary and Discussion

This case study focused on the evolution of NBF dynamics and stratification during IOP16 (26-29 October 2012), which, during

a first period (IOP16a), was characterized by weak southwesterly winds followed, in period IOP16b, by a strong northerly wind event.

Before IOP16, the light AW present south of the NBF spread up to $42°\mathrm{N}$. After IOP16, the observations showed a southward displacement of the NBF, of several tens of kilometres, with a rapid decrease in SST, larger than $4°\mathrm{C}$ in the frontal zone. To better describe and understand the NBF dynamics during this case-study we performed an air-sea coupled simulation

at kilometric scale. This simulation is in good agreement with the observations and has the ability to reproduce meso and submesoscale structures (eddies, fronts, filaments). The simulation showed that, before the strong northerly wind event, the NBF reached the Lion buoy and surface filaments of light water (not visible in the satellite SST) became detached and were entrained northward. During the strong wind event, the simulation reproduced the rapid displacement of NBF, of several tens of kilometres, and the SST decrease by $4°\mathrm{C}$ at the Lion buoy. Furthermore, the filaments were dissipated. As suggested by

Estournel et al. (2016a), the SST decrease at the Lion buoy was associated with the NBF dynamics. This dynamics also impacted the oceanic stratification with a significant loss (gain) of about $40 \ \mathrm{kg \ m^{-2}}$ ($20 \ \mathrm{kg \ m^{-2}}$) of stratification along the eastern (western) part of the NBF meander.





During the period of light southwesterly wind, the NBF clearly impacted the wind stress and the net heat flux through two effects: first a decrease of the air-sea exchanges on the cold side of the NBF (to the north), and second, as explained by Chelton and Xie (2010), the formation of a wind stress curl.

During the strong wind event, contrary to the low wind event, the NBF had little impact on the wind stress distribution. The

wind stress curl was mainly linked to the channelling of the Tramontane and Mistral by the continental orography. In contrast, the NBF directly impacted the heat and water budgets. The rapid SST cooling induced significant air-sea coupling, as suggested in other case studies in the NWMS (Lebeaupin Brossier and Drobinski, 2009; Small et al., 2008, 2012; Renault et al., 2012; Rainaud et al., 2017), by stabilizing the unstable atmospheric boundary layer and leading to a decrease of the turbulent fluxes. Furthermore, this study pointed out that the surface cooling was partly associated with a rapid NBF displacement, which led to

additional air-sea coupling feedback. Therefore, a good representation of the frontal dynamics is essential to correctly depict the space and time evolution of the air-sea exchanges.

Finally, a stratification budget diagnosis was performed during the low and strong wind periods. The results differed greatly according to the wind regime. During the low wind period, the evolution of the stratification in the frontal zone was directly controlled by the advective processes whereas, during the strong wind period, it was controlled by the Ekman buoyancy flux,

which could be up to three times stronger than the diabatic heat flux in the eastern part of the NBF. This flux directly impacted the frontal dynamics and the stratification variations. The stratification increased by up to $20 \, \text{kg m}^{-2}$ in the western part of the NBF meander (where the wind was parallel to and in the same direction as the current) and decreased by up to $40 \, \text{kg m}^{-2}$ in the eastern part of the NBF meander (where the wind was parallel but in the opposite direction to the current). The advective processes also played a role in the evolution of stratification. One of these processes clearly appears to be the Ekman pumping.

Finally, the inertial waves also impacted the stratification, with variations of the order of $10 \, \text{kg m}^{-2}$.

To the best of our knowledge, this study is the first realistic study that uses an air-sea coupled model to evaluate the impact of Ekman buoyancy fluxes on an oceanic front that is not topographically controlled. Without bathymetric constraint, the front moves by several tens of kilometres during the strong wind events. This displacement is associated with a marked loss of buoyancy and a rapid destratification, largely induced by the wind/front interaction. In an academic study, Thomas (2005)

described a frontogenesis process generated by the Ekman buoyancy flux. This process is not reproduced in our real case study. On the contrary, after the strong wind event, the front is less marked (Fig. 7). The front displacement probably prevents this process. In our case study the frontal dynamics clearly appears to be an air-sea coupling process. However, the impact of this coupling depends directly on the wind regime, as suggested by Small et al. (2008). During low wind periods, the NBF affects the diabatic buoyancy flux and the interactions between the thermal front and the Ekman pumping generated by wind stress,

and can also impact the Ekman buoyancy flux. This interaction is, therefore, air-sea thermal and dynamic coupling. During strong wind periods, the NBF position and dynamics clearly affect the diabatic buoyancy flux. They also affect the wind stress, but only weakly. There is, therefore, mainly air-sea thermal coupling.

The diabatic buoyancy flux and the adiabatic Ekman buoyancy flux are relevant parameters to quantify the thermal and the dynamic air-sea coupling effects on the evolution of stratification. However, this study also shows that the advective processes

have an important effect on the stratification evolution. In a future work, the various sources of advection could be distinguished



in the stratification budget to better understand the coupling effects, particularly the Ekman pumping. The interaction between inertial waves and front could also be studied in detail. For example, (Thomas et al., 2016) show that the Ekman buoyancy flux is more efficient when the inertial waves reduce the stratification.

This case-study of the North Balearic Front highlights the major role of Ekman buoyancy fluxes in frontal dynamics and
stratification evolution. The Ekman buoyancy fluxes could significantly influence the open-ocean deep convection that occurs in the centre of the cyclonic gyre of the Gulf of Lion during winter as shown by Giordani et al. (2016), who highlight a large impact along the eastern and northern branches of the NC. Estournel et al. (2016a) suggest that the autumnal NBF dynamics could play a major role in the preconditioning of the deep convection. Our study shows that this process is principally driven by Ekman buoyancy fluxes. In a future work, the impact of the Ekman buoyancy flux, all around the cyclonic gyre circulation,
could be quantified during the different phases of the open-ocean deep convection.

*Acknowledgements.* This work is a contribution to the MISTRALS/HyMeX programme through the ASICS-MED (ANR-12-BS06-0003) project funded by the French National Agency for Research (ANR). Data were obtained from the HyMeX programme, sponsored by grants from MISTRALS/HyMeX and Météo-France. The authors acknowledge the international ARGO programme, the LEFE/GMMC programme and the French NAOS project for supporting the deployment of profilers. Argo and CTD data were collected and made freely available by the
CORIOLIS project (http://www.coriolis.eu.org) and programmes that contribute to it. We acknowledge the crews of R/V Suroit and Tethys II and the scientists involved in the different cruises mentioned in this paper. Numerical simulations were performed using HPC resources from CALMIP (CALcul en MIdi-Pyrénées, projects 1247, 09115 and 1325) and GENCI (Grand Equipement National de Calcul Intensif, project 010569)





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





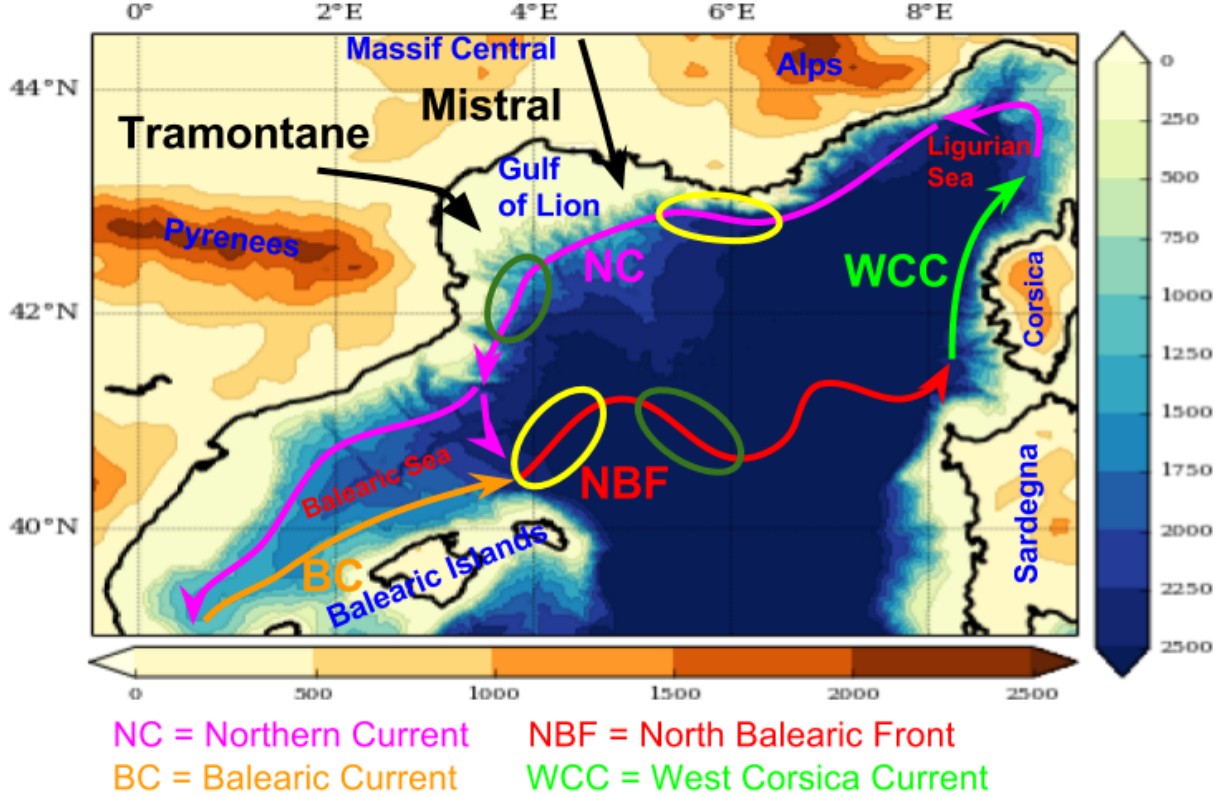

**Figure 1.** Schematic representation of the surface oceanic circulation in the North Western Mediterranean Sea. Coloured area: orography in brown and bathymetry in blue. The directions of the prevailing Mistral and Tramontane winds are indicated with black arrows. Up-front and down-front wind zones are indicated with yellow and green ellipses respectively.





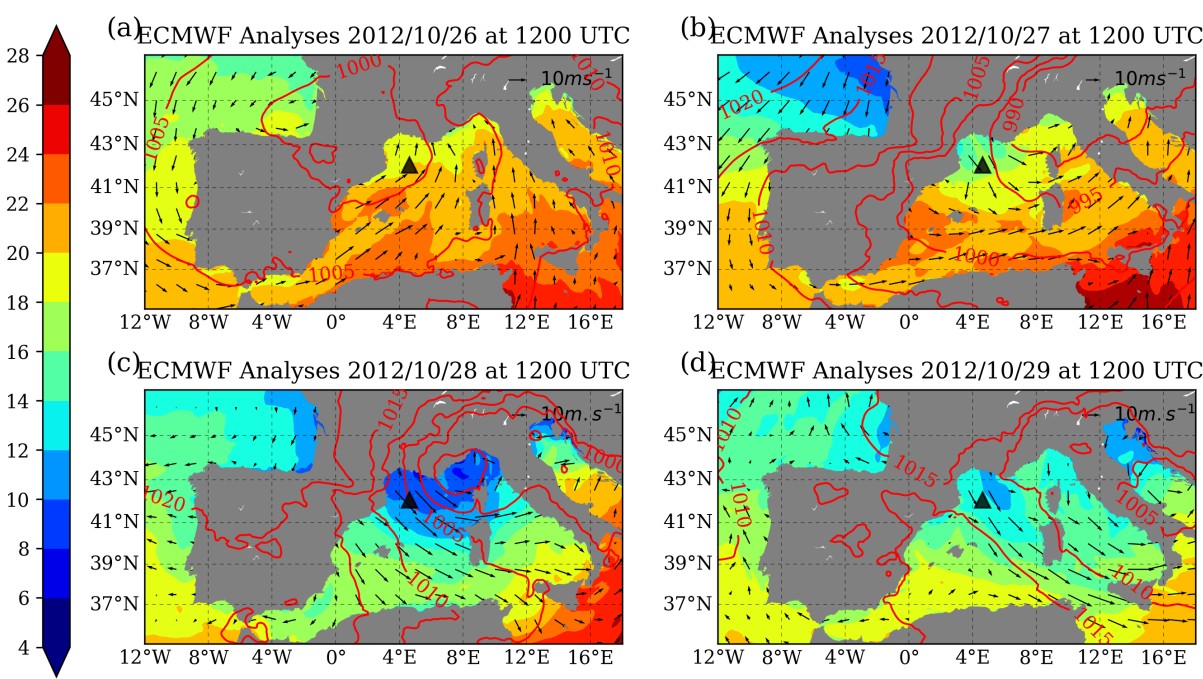

**Figure 2.** Atmospheric surface synoptic conditions from ECMWF analyses at 1200 UTC (a) on 2012/10/26, (b) on 2012/10/27, (c) on 2012/10/28 and (d) on 2012/10/29. Coloured area: atmospheric surface temperature in °C. Contour lines: mean sea level pressure in hPa. Arrows: surface wind. The black triangle indicates the position of the Lion meteorological buoy.





**Figure 3.** Time series at the Lion meteorological buoy of (a) the 10 m wind speed, (b) the 2 m air temperature, (c) the 2 m air humidity and (d) the Sea Surface Temperature. Observations in black and simulations in blue.





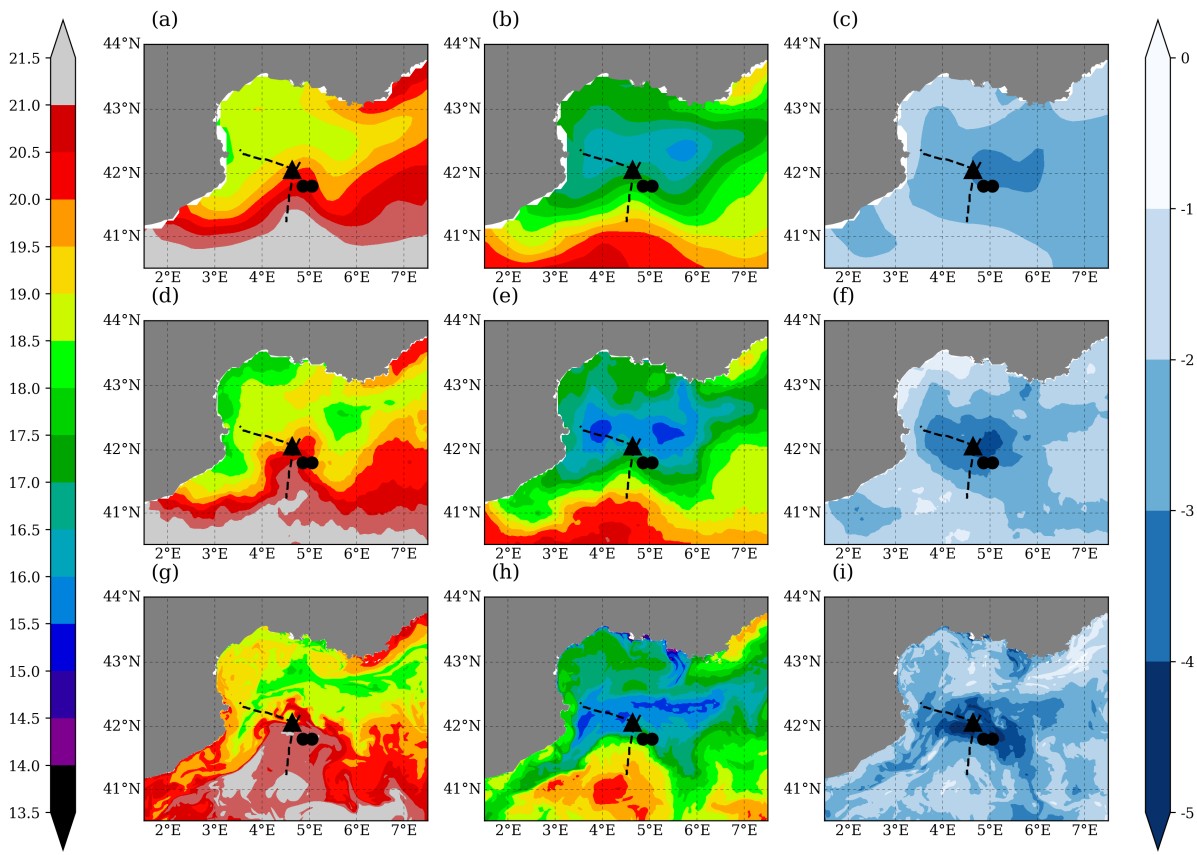

**Figure 4.** Sea Surface Temperature analysis from the OSTIA product (a-c), from the Mediterranean Copernicus product (d-f), and from the coupled simulation (g-i) at 0000 UTC on 25 October (a, d and g), at 0000 UTC on 30 October (b, e and h) and difference between 0000 UTC on 30 October and on 25 October (c, f and i). The triangle indicates the Lion buoy. The black dots correspond to the position of the ARGO profiles shown in Fig. 6 and the dashed lines to the glider sections shown in Fig. 5.





**Figure 5.** SST measurement (a), simulation (b) and bias of the simulation (c), along the trajectory of the glider Eudoxus (moving westward) and Campe (moving southward). Vertical section of potential temperature between 0 and 200 m depth for the Eudoxus glider observation (d) and simulation (e) and the Campe glider observation (f) and simulation (g).





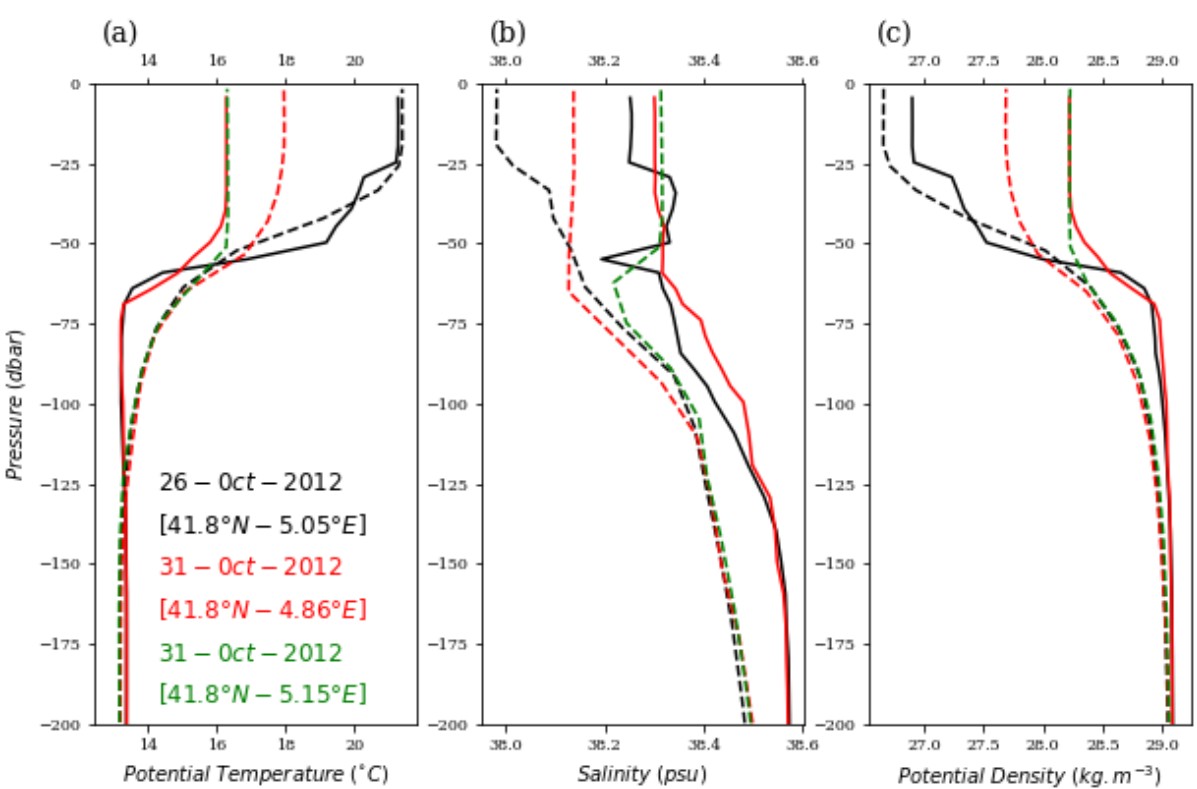

**Figure 6.** Argo potential temperature (a), salinity (b) and potential density (c) measured (solid line) and simulated (dashed line) at the different dates indicated in the figure. For 31 October, only simulated profiles are given (in green).





**Figure 7.** Sea surface density in kg m$^{-3}$ (left column) and Stratification Index at 250 m in kg m$^{-2}$ (right column), at 0000 UTC on 26 October (a and d), at 0000 UTC on 30 October (b and e) and difference between 0000 UTC on 30 October and on 26 October (c and f). The black contour (a and c) represents the 27.0 kg m$^{-3}$ isopycnal, the blue contour (b and c) represents the 28.0 kg m$^{-3}$ isopycnal and the red line (a-c) represents the surface density gradient greater than 0.1 kg m$^{-3}$ per kilometre. The black straight lines indicate the position of the vertical sections shown in Fig. 8.





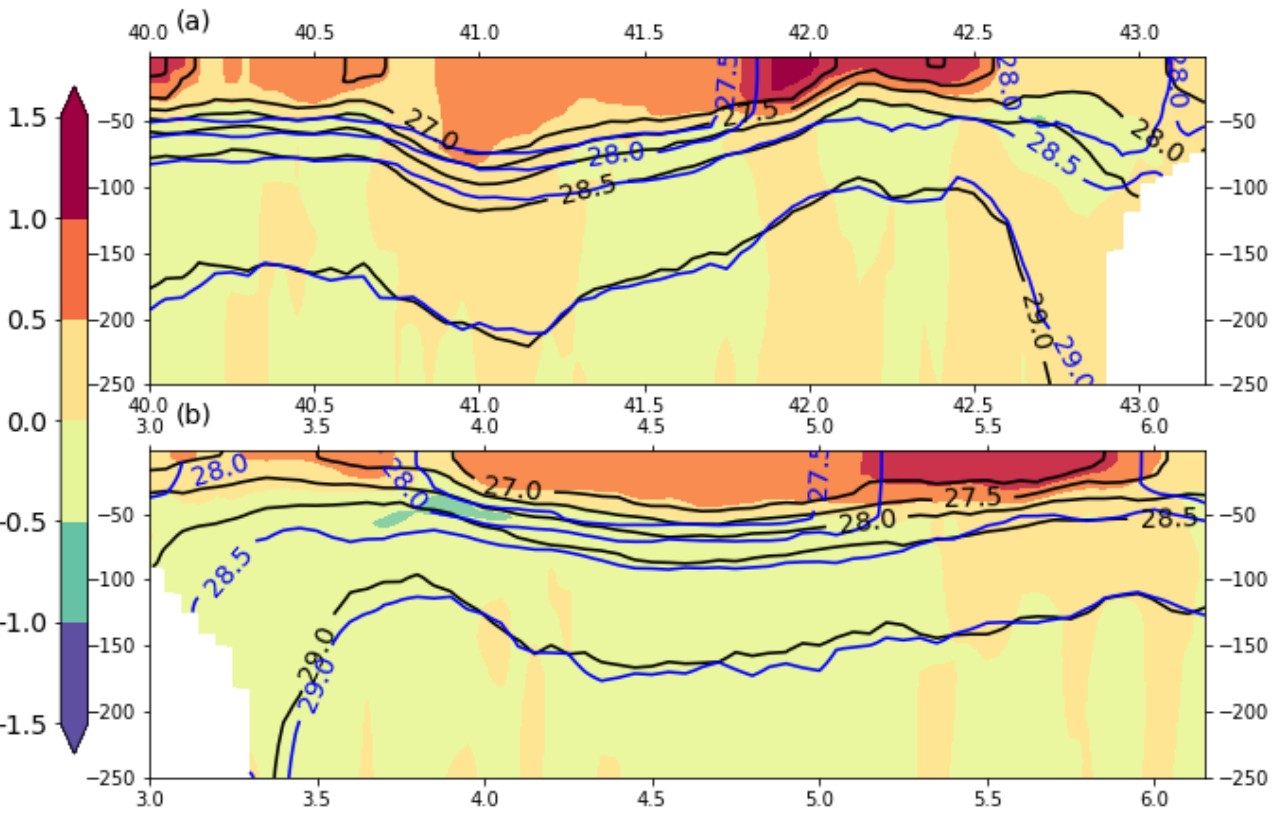

**Figure 8.** (a) South-North section at 4.65°E and (b) West-East section at 41.7°N (see Fig. 7 for their positions) of the potential density difference between 0000 UTC on 30 October and 0000 UTC on 25 October. Contour lines: Isopycnals on 25 October 2012 (black lines) and 30 October 2012 (blue lines).





**Figure 9.** Time series of (a) the wind stress, (b) the net heat flux, and (c) the water flux simulated at the Lion meteorological buoy.





**Figure 10.** (a and b) wind stress (N m$^{-2}$), (c and d) net surface heat flux (W m$^{-2}$), (e and f) surface water flux (mm day$^{-1}$) averaged over IOP16a (left column) and IOP16b (right column). Contour lines: sea surface density averaged over IOP16a (left) and IOP16b (right), contour interval: 0.3 kg m$^{-3}$.





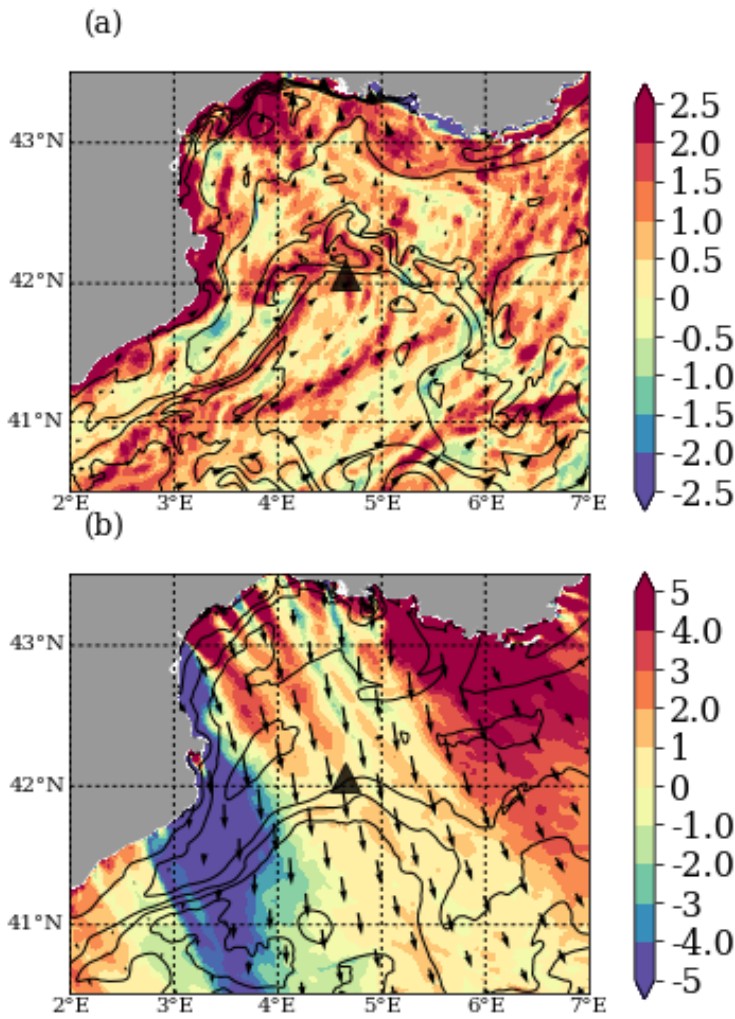

**Figure 11.** Wind stress curl (N m$^{-3}$×10$^6$) averaged over IOP16a (top) and IOP16b (bottom). Contour lines: sea surface density averaged over IOP16a (left) and IOP16b (right), contour interval: 0.3 kg m$^{-3}$.





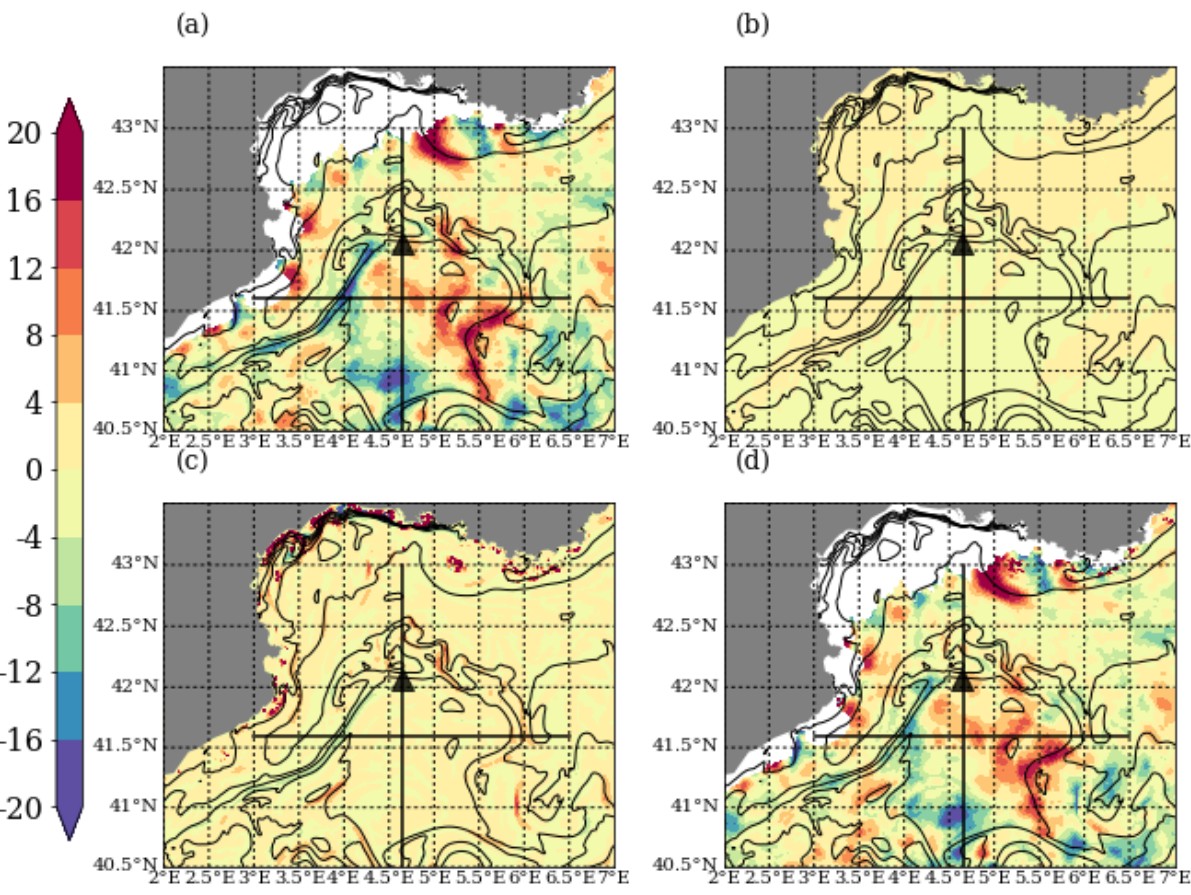

**Figure 12.** Stratification variation budget diagnosis (Eq. 5) during IOP16a, (a) SI difference, (b) diabatic buoyancy mass flux, (c) Ekman buoyancy mass flux, and (d) residual term.





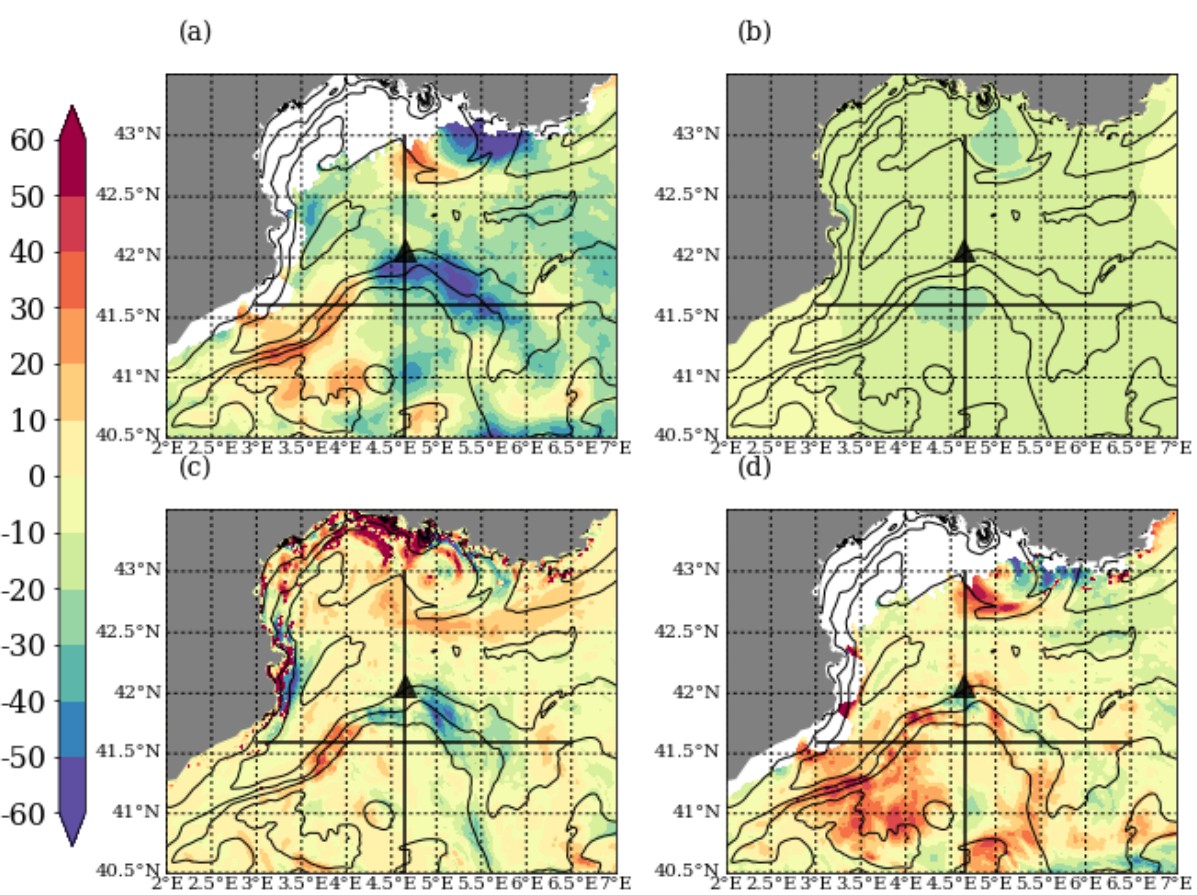

**Figure 13.** Same figure as Fig. 12 for IOP16b.


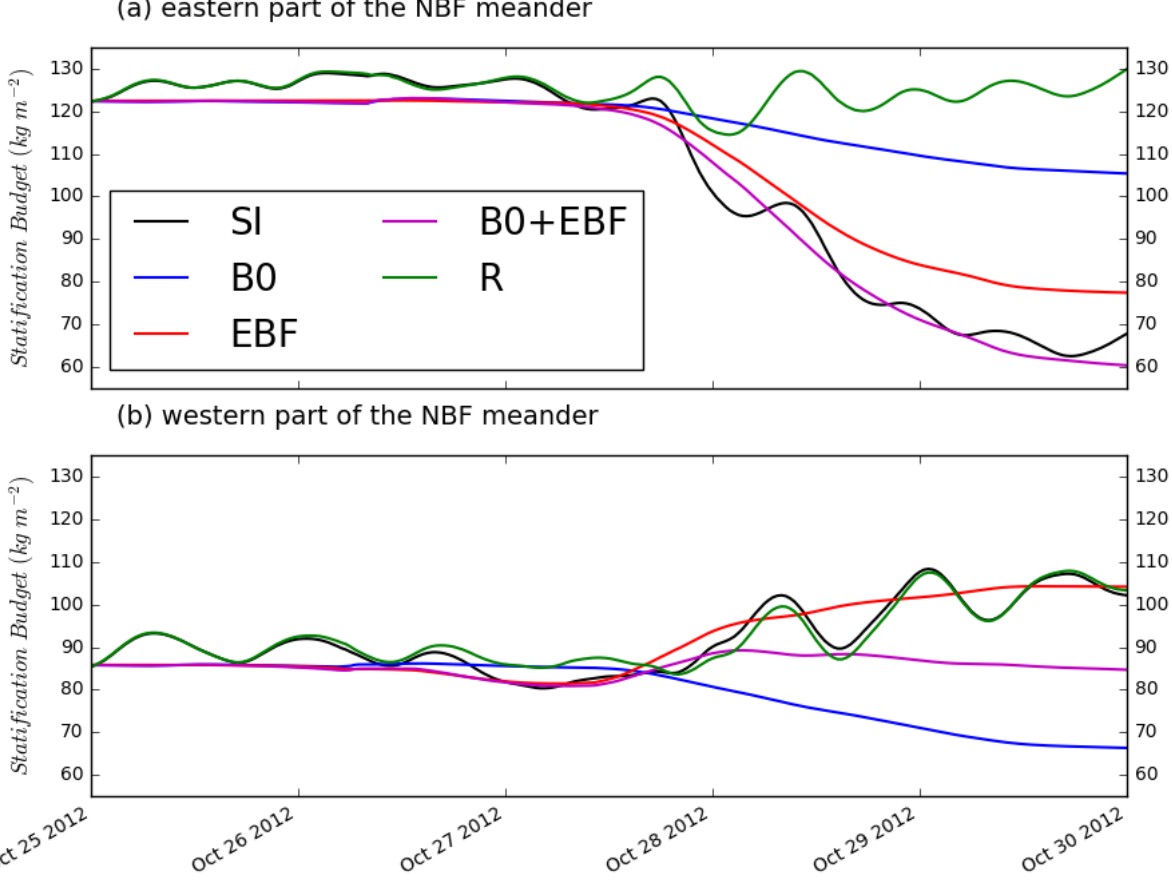

**Figure 14.** Time series of the different terms of the stratification budget diagnosis (SI in black, diabatic buoyancy flux in blue, Ekman buoyancy flux in red, total buoyancy flux in magenta and residual in green): (a) for the eastern part of the NBF meander (averaged over 10 km$^2$ around 41.9°N and 5.1°E) and (b) for the western part of the NBF meander (averaged over 10 km$^2$ around 41.6°N and 3.9°E). In order to compare the different terms, all the curves start with the same value, which is the initial SI.





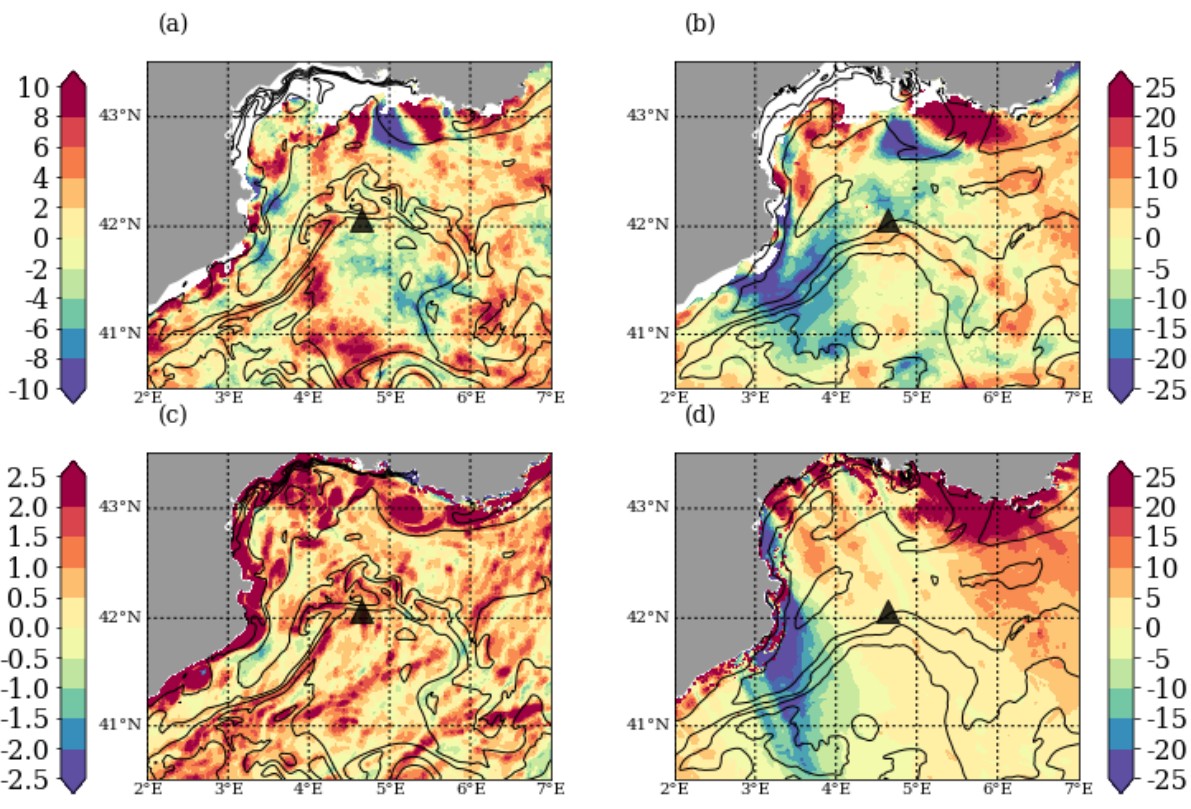

**Figure 15.** (a and b) 28.5 kg m$^{-3}$ isopycnal deepening (m) and (c and d) Ekman pumping (m) during IOP16a (left column) and IOP16b (right column). Contour lines: sea surface density averaged over IOP16a (left) and IOP16b (right). Contour interval: kg m$^{-3}$.