# Peer review of "Dynamics of North Balearic Front during an autumn Tramontane and Mistral storm: air-sea coupling processes and stratification budget diagnostic"

_Ocean Science, 2018_

## Referee Comment (RC1) · Anonymous Referee #1 · 9 Apr 2018

**Review Seyfried et al. 2018**

The Northwestern Mediterranean is an important region for the Mediterranean Overturning Circulation through the formation of Western Mediterranean Deep Water. The authors provide an original study of the north Balearic front, bordering the open-ocean deep convection area of the Western Mediterranean. They combine observations and a realistic air-sea coupled model to analyse the thermal and dynamical air-sea coupling for different wind regimes in the north Balearic front region. Particularly, they highlight the importance of the adiabatic Ekman buoyancy flux in destratifying the frontal zone during strong wind event (three times greater than the air-sea buoyancy flux).

I really enjoyed reading this manuscript which I think was clearly written and well structured. I was particularly amazed by the excellent agreement between the coupled model system and the met-ocean observations.

I think the work is significant and deserve prompt publication. I have however a few comments I would like the authors to address before the manuscript is considered for publication.

**Minor comments:**

p5,I 12: By looking fig 6c), the MLD doesn't seem to deepen only by a few meters, it looks like it deepened from ~25m to ~40m. To avoid this interpretation problem, you should indicate the MLD on the different profiles of the figure, particularly because you mentioned it in the text.

P5, I14: slight is not very quantitative can you be more accurate?

P7, I3-4: Any explanation why these small-scale structures were not observed?

P7, I8: can you quantify how much smaller (10%, 100%,...)?

p7, I 10-11: What about the large 0.2 bias between model and obs in Oct. 26? Is it also due to the fact that the modelled profile was not in the same region/regime? Adding small maps on fig6, with the modelled and observed SST, zooming on the location of the profiles, would help understand if the profile were taken from the same type of region.

P9, I2: "the density does not increase"  $\rightarrow$  the density appears to increase when looking the position of the isopycnal and the colorbar

p9, I11: when you speak about correlation you should give the correlation coefficient in the text, otherwise you should find another term.

P10, I22: E and P are not variables shown in the equation E (4). You should rather introduce Qnet and Fw.

P11, I2: Can you give the origin or the kind of processes which are part of R? Later on, you speak about advective processes, you should also introduce it here.

P11, I15-16: you could precise that the stratification fain is of the same order of magnitue than the cumulated EBF effect.

P12, I14: in the BNF zone  $\rightarrow$  in the eastern part of the NBF (down-front wind)

p13, l25: "This process is not reproduced in our real case study."  $\rightarrow$  Do you have an explanation why?

Typos/ spelling mistakes:

p1,I5: ... modelling system and focused...  $\rightarrow$  modelling system. We focused

p1,I9: I will split the sentence I8-I10 in two, I found it hard to follow.

p4, I19: '..., the horizontal resolution of which is about'  $\rightarrow$ ..., with an horizontal resolution of

p6, l12: (of all the  $\ldots \rightarrow$  from all the ?

P7, I4: marked  $\rightarrow$  intense?

P7, I7: was shown by the model on October 26th.

P7, I8: ... an increase in salinity and density [...], except for salinity.  $\rightarrow$  something is not clear, can you double check the meaning? P7, I15: obviously  $\rightarrow$  slightly?

P7, I22: red  $\rightarrow$  yellow-red?

P7, I28: less strong  $\rightarrow$  reduced?

P8, I17: and after IOP16 (blue lines in fig 8)

p8, I27: The EW section (Fig. 8.b) before IOP16, which crosses the meander (Fig. 7c), intersects the NBF  $\rightarrow$  The EW section (Fig. 8.b) crosses the meander (Fig. 7c) before IOP16. It intersects the NBF

p9, I1: in the heart  $\rightarrow$  in the core?

**Figures:**

fig3: You should add the ticks for all the time mentioned in the text (e.g. 18 UTC on October 27). Did you display the ticks for 0000UTC or 1200UTC?

fig4. Drawing the NBF position on figure 4 would help to follow the text (p7,I9)

fig5: - Can you add an indication of the period of strong wind mentioned in the text (e.g. by an horizontal line on top of the sections?)

- The tick labels cannot be seen clearly. Can you make sure to make them visible? You should also plot the MLD on figure (d)-(g) as you discuss it in the text, but it's difficult to locate it on the suplots.

Fig6: You should need the MLD on all profiles and add the criterion used (density based?), for example by a small horizontal line on obs and model profiles.

It's difficult to clearly localised the green dash profile location on figure 4. Could you a subplot where you indicate the position of argo profiles in Oct. 26th for both model and obs SST maps? (by zooming on a box such as 42.5N-42.2N / 4.5E-5.5E). Can you also do it for the locations of the 2 profiles taken in October 31st? It could help understanding why the profiles doesn't exactly match.

Fig7: - I4: line (a-c)  $\rightarrow$  do you mean (a-b)?

- You should keep a numbering consistent between all figures (increasing from left to right and top to bottom)

Fig8: This figure is difficult to read.

- For clarity, you should leave more horizontal and vertical spaces between the colorbar and vertical sections.

- Can you add axe labels for plots and colorbar?

- Do we need to see all these isopycnals? The superposition of isopycnal doesn't make the figure easy to read. What about keeping the 27 and 28 isopycnals as plain lines and the other as small dash lines? Could you also make sure that the contour labels are visible? (by adding them manually in the middle of the contour line for example?) Can you indicate the surface position of the front on the 25/10 and on 30/11 on top of the section by a specific marker (e.g. reverse triangle, arrow, ...)? You could indicate the cyclonic gyre extension you are mentioning in the text.

Fig 14: Could you indicate, on figure 13 for example, the area chosen for the two regions a and b?

---

## Referee Comment (RC2) · Anonymous Referee #2 · 8 Jun 2018

The paper presents an original study of the North Balearic Front dynamics during a strong wind event, combining observations, high-resolution coupled modelling and stratification budget. The authors give a complete overview of the atmospheric event and of the ocean front evolution. The validation of the coupled simulation is very convincing. The use of a stratification budget seems very promising in particular to better understand the coupled mechanisms involved.

Nevertheless, the fact that the equations are not fully detailed and that the residual term is not described but finally appears as a dominant term, gives the impression that

only a part of this budget is considered.

Consequently, I suggest some major revisions to improve the paper before accepting its publication.

- - -

So, my main comment concerns the stratification budget (section 6) which is very briefly presented:

First, it could be very helpful to give a physical view of EBF. Reading Thomas and Lee (2005), I understood it is a destabilizing flux leading to convection and frontal intensification, but clearly I am not sure about my interpretation...

p10, line 10: "the friction induced a wind-driven or Ekman Buoyancy Flux (EBF) given by:"

Where does equation 6 come from? There is no residual term in equation 13 of Thomas and Lee (2005)? What is the value of H considered? Is the equation valid for any H or is there a limit considering the depth of the Ekman layer?

Maybe, one possibility to clarify this section is to give an enlarged description of the reasoning in an annexe. Obviously, the idea is not to reproduce the work from Thomas and Lee (2005) but to try to give the main insights.

Concerning the residual term, it appears later in the text that you can attribute its value to horizontal advection or vertical advection/Ekman pumping. How? If there is a way, you must extract these terms from R and plot their values to complete the stratification budget.

Finally, in section 7, it could be also interesting to discuss the possible limitations due to the hydrostatic assumption or to the convection parametrization in the ocean model.

OSD
Other comments:

p2, lines 2-9: There is a kind of mixing between front/current in the very first sentence of the introduction. In my opinion, it could be helpful to describe the cyclonic circulation with the various branches/currents and then how it constrains/forms the surface density gradient...?

p2, line 21: "maintained strong precipitation offshore and over the southeastern French coasts"

p2, lines 30-32: I am curious to know if there is any action of the perpendicular wind component?

p4, lines 20-21: It appears from fig 4 that the effective resolution of the OSTIA and Copernicus products are much larger than the indicated resolution (6 and 1 km, resp.). This is somehow mentioned p6, line 25 ("The horizontal resolution of the latter...of the model."), but could be rapidly indicated in this section 2.2 or commented.

p6, line 34: replace "extremity" by "end".

p8, lines 4-5 should not be in italic.

p8, line 10: "Modified": This is the first time this term appears. Could you explain it?

p9, line 16: Remove here the comment about density: "During strong wind event, evaporation dominated and led to an upward (positive) water flux."

p9, line 19: You may refer here to fig 2 (instead of fig 10a)?

p10: See my main comment + Please, detail g,  $rho_0$ , Qnet and Fw; Some sentences should not be in italic.

p12, line 14: "during IOP16b"

p13, line 26: "the front is less marked"  $\rightarrow$  this is not so clear for me from figs 4 and 7. Are you talking about a smoothing? a reduction of the temperature difference?
p14, line 2: "For example, Thomas et al. (2016) show"

p15: refs Drobinski et al. and Ducrocq et al.  $\rightarrow$  2014; In addition, I think URLs in this "References" section must refer to the "doi.org" pages.

Figure 5: precise "observation" or "simulation" on each panel if possible. For (c), the caption tells it is the simulation bias but it seems to be more 'glider minus simulation'...

Figure 7c: The red contour is not visible.

Figure 8 is difficult to read. Please consider here to plot separately the density sections for 25 oct., 30 oct. and the differences.

Figure 10: Please, improve the resolution in order to distinguish the arrows. Precise in the caption if the arrows are for the wind or the wind stress + a reference length for vectors must be added.

Figure 14: the plots of "B0+EBF" are not necessary...

---

## Author Comment (AC1) · 6 Sep 2018

Response to Referee #1

Authors' Answer: The paper has been revised according to the comments from the reviewers and we thank both reviewers for their very helpful comments and suggestions. Our point-by-point response is inserted in the reviewer's comments.

Reviewer's Comment: The Northwestern Mediterranean is an important region for the Mediterranean Overturning Circulation through the formation of Western Mediter-

ranean Deep Water. The authors provide an original study of the north Balearic front, bordering the open-ocean deep convection area of the Western Mediterranean. They combine observations and a realistic air-sea coupled model to analyse the thermal and dynamical air-sea coupling for different wind regimes in the north Balearic front region. Particularly, they highlight the importance of the adiabatic Ekman buoyancy flux in destratifying the frontal zone during strong wind event (three times greater than the air-sea buoyancy flux). I really enjoyed reading this manuscript which I think was clearly written and well structured. I was particularly amazed by the excellent agreement between the coupled model system and the met-ocean observations. I think the work is significant and deserve prompt publication. I have however a few comments I would like the authors to address before the manuscript is considered for publication.

Minor comments:

RC: p5,l 12: By looking fig 6c), the MLD doesn't seem to deepen only by a few meters, it looks like it deepened from ∼25m to ∼40m. To avoid this interpretation problem, you should indicate the MLD on the different profiles of the figure, particularly because you mentioned it in the text.

AA: The MLD have been added to the Fig 6c. The MLD deepen from ∼25 to ∼40m. The text has been modified accordingly.

RC: P5, l14: slight is not very quantitative can you be more accurate?

AA: "slight" has been replaced by a MLD deepening of about 20 meters.

RC: P7, l3-4: Any explanation why these small-scale structures were not observed?

AA: The filament structure simulated (see Fig. 4g) is not positioned at the same place in the reality.

RC: P7, l8: can you quantify how much smaller (10%, 100%,...)? p7, l 10-11: What about the large 0.2 bias between model and obs in Oct. 26? Is it also due to the fact that the modelled profile was not in the same region/regime? Adding small maps

on fig6, with the modelled and observed SST, zooming on the location of the profiles, would help understand if the profile were taken from the same type of region.

AA: This paragraph p7 l7-11 was modified : "At the end of the strong wind event, the simulation showed a decrease of temperature and an increase in salinity, density and MLD. The temperature decrease was smaller than in the observations, 3.3°C compared to 5.3°C. Whereas the salinity increase was more important in the simulation than in the observations, 0.15 compared to 0.05.The increase of density and MLD were smaller than in the observations, 1 km m-3 and 15 m compared to 1.32 kg m-3 and 25 m, respectively. However, it is worth to note that the observed profile is located to the north of the NBF whereas the corresponding one in the simulation is within the frontal zone. When the comparison is made with a simulated profil located slightly further east (41.8°N -5.15°E, green dashed line in Fig. 6), the simulation results are much closer to the Argo observations." The Argo profiles positions are now available on Figure 4.

RC: P9, l2: "the density does not increase" → the density appears to increase when looking the position of the isopycnal and the colorbar.

AA: Modified

RC: p9, l11: when you speak about correlation you should give the correlation coefficient in the text, otherwise you should find another term.

AA: Modified

RC: P10, l22: E and P are not variables shown in the equation E (4). You should rather introduce Qnet and Fw.

AA: Done

RC: P11, l2: Can you give the origin or the kind of processes which are part of R? Later on, you speak about advective processes, you should also introduce it here.

AA: The origin of processes which are part of R are now introduced p10 l23: "The stratification index variation at depth H (with H =250 m, H>MLD) between times T1 and T2 can be approximated by the integral of buoyancy mass flux between times T1 and T2 (Eq 5). In order to evaluate the competing roles of the diabatic and Ekman buoyancy fluxes on stratification variation, these two term are diagnosed and compared to the stratification variations. Finally, to close our stratification budget diagnosis we evaluate the residual term corresponding to other potential sources of horizontal and vertical advection of buoyancy (geostrophic circulation, frontogenesis, Ekman pumping, ...) that are not directly diagnosed in this study".

RC: P11, l15-16: you could precise that the stratification fain is of the same order of magnitude than the cumulated EBF effect.

AA: Done

RC: P12, l14: in the BNF zone → in the eastern part of the NBF (down-front wind)

AA: Done

RC: p13, l25: "This process is not reproduced in our real case study." → Do you have an explanation why?

AA: We explain that by the fact that the front is not constrained by the bathymetry.

Typos/ spelling mistakes:

RC: p1,l5: ... modelling system and focused... → modelling system. We focused

AA: Done

RC: p1,l9: I will split the sentence l8-l10 in two, I found it hard to follow.

AA: Done

RC: p4, l19: '..., the horizontal resolution of which is about' →..., with an horizontal resolution of

AA: Done

RC: p6, l12: (of all the .... → from all the ?

AA: Done

RC: P7, l4: marked → intense?

AA: Modified

RC: P7, l7: was shown by the model on October 26th

AA: Corrected

RC: P7, l8: ... an increase in salinity and density [...], except for salinity. → something is not clear, can you double check the meaning?

AA: This section has been modified.

RC: P7, l15: obviously → slightly?

AA: Modified

RC: P7, l22: red → yellow-red?

AA: Modified

RC: P7, l28: less strong → reduced?

AA: Modified

RC: P8, l17: and after IOP16 (blue lines in fig 8)

AA: Modified

RC: p8, l27: The EW section (Fig. 8.b) before IOP16, which crosses the meander (Fig. 7c), intersects the NBF → The EW section (Fig. 8.b) crosses the meander (Fig. 7c) before IOP16. It intersects the NBF

AA: Modified

RC: p9, l1: in the heart → in the core?

AA: Corrected

Figures: RC: fig3: You should add the ticks for all the time mentioned in the text (e.g. 18 UTC on October 27). Did you display the ticks for 0000UTC or 1200UTC?

AA: The figure 3 has been modified to be more readable.

RC: fig4. Drawing the NBF position on figure 4 would help to follow the text (p7,l9)

AA: Done

RC: fig5: - Can you add an indication of the period of strong wind mentioned in the text (e.g. by an horizontal line on top of the sections?) - The tick labels cannot be seen clearly. Can you make sure to make them visible? You should also plot the MLD on figure (d)-(g) as you discuss it in the text, but it's difficult to locate it on the suplots.

AA: The figure 5 has been modified to be more readable.

RC: Fig6: You should need the MLD on all profiles and add the criterion used (density based?), for example by a small horizontal line on obs and model profiles.

AA: The MLD for all profiles was added to the density profile.

RC: It's difficult to clearly localised the green dash profile location on figure 4. Could you a subplot where you indicate the position of argo profiles in Oct. 26th for both model and obs SST maps? (by zooming on a box such as 42.5N-42.2N / 4.5E-5.5E). Can you also do it for the locations of the 2 profiles taken in October 31 st? It could help understanding why the profiles doesn't exactly match.

AA: The dash profile location has been added on figure 4.

RC: Fig7: - l4: line (a-c) → do you mean (a-b)? - You should keep a numbering consistent between all figures (increasing from left to right and top to bottom)

AA: Done

RC: Fig8: This figure is difficult to read. - For clarity, you should leave more horizontal and vertical spaces between the colorbar and vertical sections. - Can you add axe labels for plots and colorbar? - Do we need to see all these isopycnals? The superposition of isopycnal doesn't make the figure easy to read. What about keeping the 27 and 28 isopycnals as plain lines and the other as small dash lines? Could you also make sure that the contour labels are visible? (by adding them manually in the middle of the contour line for example?) Can you indicate the surface position of the front on the 25/10 and on 30/11 on top of the section by a specific marker (e.g. reverse triangle, arrow, ...) ? You could indicate the cyclonic gyre extension you are mentioning in the text.

AA: The figure 8 has been modified to be more readable.

RC: Fig 14: Could you indicate, on figure 13 for example, the area chosen for the two regions a and b?

AA: Done

**Fig. 1.** Time series at the Lion meteorological buoy of (a) the 10 m wind speed, (b) the 2 m air temperature, (c) the 2 m air humidity and (d) the Sea Surface Temperature. Observations in black and simulations

**Fig. 2.** Sea Surface Temperature analysis from the OSTIA product (a-c), from the Mediter-
ranean Copernicus product (d-f), and from the coupled simulation (g-i) at 0000 UTC on 25
October (a, d and g), at 0000 UT

**Fig. 3.** SST measurement (a), simulation (b) and bias of the simulation (c), along the trajectory of the glider Eudoxus (moving westward) and Campe (moving southward). Vertical section of potential temperature

(a)

(b)

(c)

26 − 0ct − 2012
[41.8° N − 5.05° E]

31 − 0ct − 2012
[41.8° N − 4.86° E]

31 − 0ct − 2012
[41.8° N − 5.15° E]

*Pressure (dbar)*

*Potential Temperature (°C)*

*Salinity (psu)*

*Potential Density (kg. m⁻³)*

**Fig. 4.** Argo potential temperature (a), salinity (b) and potential density (c) measured (solid line) and simulated (dashed line) at the different dates indicated in the figure. For 31 October, only simulated

**Fig. 5.** Sea surface density in \unit{kg\ m^{-3}} (left column) and Stratification Index at 250 m in \unit{kg\ m^{-2}} (right column), at 0000 UTC on 26 October (a and d), at 0000 UTC on 30 October (b and e)

**Fig. 6.** (a) South-North section at 4.65\unit{ˆ\circ E} and (b) West-East section at 41.7\unit{ˆ\circ N} (see Fig. \ref{Fig7} for their positions) of the potential density difference between 0000 UTC on 30 Oct

(a)

(b)

(c)

(d)

**Fig. 7.** Same figure as Fig. 12 for IOP16b.

---

## Author Comment (AC2) · 6 Sep 2018

Response to Referee 2

Authors' Answer: The paper has been revised according to the comments from the reviewers and we thank both reviewers for their very helpful comments and suggestions. Our point-by-point response is inserted in the reviewer's comments.

Reviewer's Comment: The paper presents an original study of the North Balearic Front dynamics during a strong wind event, combining observations, high-resolution coupled

modelling and stratification budget. The authors give a complete overview of the atmospheric event and of the ocean front evolution. The validation of the coupled simulation is very convincing. The use of a stratification budget seems very promising in particular to better understand the coupled mechanisms involved. Nevertheless, the fact that the equations are not fully detailed and that the residual term is not described but finally appears as a dominant term, gives the impression that only a part of this budget is considered. Consequently, I suggest some major revisions to improve the paper before accepting its publication.

So, my main comment concerns the stratification budget (section 6) which is very briefly presented:

First, it could be very helpful to give a physical view of EBF. Reading Thomas and Lee (2005), I understood it is a destabilizing flux leading to convection and frontal intensification, but clearly I am not sure about my interpretation.

AA: Rephrased and Added sentences p10, line 9 to 23 : "The stratification variation of water column can be modified through diabatic processes and horizontal or vertical advection of buoyancy. Most models assume that the stratification budget is essentially driven by one-dimensional turbulent mixing of heat and water at air-sea interface. Following Merstens and Schott (1998) the air-sea exchanges induce a surface or diabatic buoyancy flux can be diagnosed as :

$$B_0 = g\alpha \frac{Q_{net}}{\rho_0 C_p} + g\beta SSS(E - P)$$
(1)

where  $B_0$  is the diabatic buoyancy flux in m2 s-3,  $\alpha$  the thermal expansion coefficient in K-1,  $C_p$  the specific heat capacity in J kg-1 K-1,  $\beta$  the saline contraction coefficient, SSS the sea surface salinity, E the evaporation and P the precipitations in m s-1.

This one-dimensional approximation is suitable if the ocean is horizontally homogeneous. In reality, mesoscale and submesoscale structures populate the ocean. Theses
structures are marked by horizontal buoyancy fronts. As shown by Thomas and Lee (2005) and Thomas and Ferrari (2008), the stratification can be significantly modified by interactions between these fronts and Ekman flow generated by frictional forcing. When the winds are down-front, the density advection of dense water over light water by Ekman transport destabilizes the water column and triggers convection. This process destratifies the water column by Ekman advection of buoyancy and mixing through the mixed layer. On the contrary when the winds are up-front the Ekman flow yields an Ekman advective restratification in the surface layer. Following Thomas and Taylor (2010), the frictional forcing induces a wind-driven or Ekman Buoyancy Flux (EBF) which can be diagnosed as :

$$\mathsf{EBF} = -\frac{g}{\rho_0} \vec{M_e} \cdot \vec{\nabla_h} \rho_{(z=0)} \tag{2}$$

where EBF is the Ekman Buoyancy Flux in m2 s-3,  $\vec{M}_e$  the Ekman transport (Eq. 3) in m2 s-1,  $\vec{\nabla}_h$  the horizontal gradient, gthe gravitational acceleration and  $\rho_{(z=0)}$  the surface density in kg m-3.

$$\vec{M_e} = \frac{\hat{z} \times \vec{\tau}}{\rho_0 \zeta_a} \tag{3}$$

where z is the vertical unit vector,  $\vec{\tau}$  the wind stress in N m-2, and  $\zeta_a$  the absolute vorticity in s-1."

RC: p10, line 10: "the friction induced a wind-driven or Ekman Buoyancy Flux (EBF) given by:"

AA: Modified

RC: Where does equation 6 come from? There is no residual term in equation 13 of Thomas and Lee (2005)? What is the value of H considered? Is the equation valid for any H or is there a limit considering the depth of the Ekman layer?
AA: I think you want to talk about equation 5. The variation of the water column stratification between times T1 et T2 can be approximated by the integral of the buoyancy flux that applies on the water column between time T1 and T2. As described above, two types of buoyancy flux generated by the air-sea exchanges can be distinguished: The diabatic buoyancy flux generated by the heat and water exchanges and the Ekman buoyancy flux generated by frictional forces. Equation 13 of Thomas and Lee (2005) corresponds to the sum of these two fluxes. As shown by Thomas and Ferrari (2008), diabatic processes and frictional forces are the only processes that can generate and destroy stratification. Our stratification budget diagnosis aims at assessing the competing roles of the diabatic buoyancy flux and Ekman buoyancy flux on the stratification evolution. However, the stratification of the water column can also be modified by horizontal and vertical advection of buoyancy. There is several sources of buoyancy advection in the water column (geostrophic circulation, frontogenesis, Ekman pumping, ...) which are in our budget induced in the so called residual term. The computation of these three terms (diabatic buoyancy flux, Ekman buoyancy flux and residual) provides a rough estimate of their respective contributions to the stratification evolution.

In our case we considered H = 250 m with H>MLD. Tests were performed tests with H = 1000 m and H = 1500, the results remain unchanged. In theory, the equation of Ekman buoyancy flux is valid as long as MLD » Ekman layer depth. However, Thomas and Lee (2005) show that this hypothesis is not crucial and the MLD can be of the same order of magnitude as the Ekman layer. In our case, the Ekman layer depth is about 30 m and the MLD about 50m.

To clarify Equation 5, we added this paragraph p10 l23: "The stratification index variation at depth H (with H =250 m, H>MLD) between times T1 and T2 can be approximated by the integral of buoyancy mass flux between times T1 and T2 (Eq 5). In order to evaluate the competing roles of the diabatic and Ekman buoyancy fluxes on stratification variation, these two term are diagnosed and compared to the stratification variations. Finally, to close our stratification budget diagnosis we evaluate the resid-
ual term corresponding to other potential sources of horizontal and vertical advection of buoyancy (geostrophic circulation, frontogenesis, Ekman pumping, ...) that are not directly diagnosed in this study".

RC: Maybe, one possibility to clarify this section is to give an enlarged description of the reasoning in an annexe. Obviously, the idea is not to reproduce the work from Thomas and Lee (2005) but to try to give the main insights.

AA: The work of Thomas and Lee (2005) is now more detailed in section 6 (see above).

RC: Concerning the residual term, it appears later in the text that you can attribute its value to horizontal advection or vertical advection/Ekman pumping. How? If there is a way, you must extract these terms from R and plot their values to complete the stratification budget.

AA: Extract the different terms of the residue is not the purpose of the paper which aims to compare the respective impact of diabatic processes and frictional forces on the evolution of stratification. In future work it would be interesting to calculate a buoyancy flux generated by Ekman pumping. However, this computation is not straightforward in a 3D realistic simulation.

RC: Finally, in section 7, it could be also interesting to discuss the possible limitations due to the hydrostatic assumption or to the convection parameterization in the ocean model.

AA: One limitation of turbulence parameterization is that the turbulence at surface is driven only by atmospheric forcing. Large Eddy Simulation (LES) with explicit resolution of turbulence shows that turbulence can also be generated through frontal instabilities (Thomas et al. 2013). It would be interesting to set up an realistic LES configuration in NWMS (currently too expensive) to explicitly solve turbulence and convective processes and to compare to our hydrostatic simulation.

Other comments:
RC: p2, lines 2-9: There is a kind of mixing between front/current in the very first sentence of the introduction. In my opinion, it could be helpful to describe the cyclonic circulation with the various branches/currents and then how it constrains/forms the surface density gradient?

AA: Rephrased sentences: "The surface circulation in the North Western Mediterranean Sea (NWMS) is formed by a cyclonic oceanic gyre (Fig. 1). This cyclonic gyre is closed to the north and west by the Northern Current (Millot, 1999), and to the east by the West Corsica Current (WCC) (Fig. 1). The south branch of the surface gyre is defined by a frontal zone, the so-called North Balearic Front (NBF). The NBF is an extension of the Balearic Current (BC), from the Balearic Sea to the Ligurian Sea (Font et al., 1988). This surface density front (100-200 m deep) separates the warm and fresh Atlantic Water (AW) which has recently entered the south of the basin from the colder and saltier AW present in the center of cyclonic gyre (Millot and Taupier-Letage, 2005). This front forms a "Lagrangian barrier" (Mancho et al., 2008) which plays an important role on the nutrients budget and planktonic ecosystem (Estrada et al., 1999) and marine ecosystems distributions (Gannier and Praca, 2007; Cotté et al., 2011)."

RC: p2, line 21: "maintained strong precipitation offshore and over the southeastern French coasts"

AA: Done

RC: p2, lines 30-32: I am curious to know if there is any action of the perpendicular wind component?

AA: There is a perpendicular wind component but we have not studied its impact.

RC: p4, lines 20-21: It appears from fig 4 that the effective resolution of the OSTIA and Copernicus products are much larger than the indicated resolution (6 and 1 km, resp.). This is somehow mentioned p6, line 25 ("The horizontal resolution of the latter...of the model."), but could be rapidly indicated in this section 2.2 or commented.
AA: Rephrased and added sentences: "The first analysis is the global OSTIA product (Donlon et al., 2012), the horizontal resolution of which is about 6 km. This product is used in the ECMWF operational model. The second analysis is the Mediterranean Copernicus product (Buongiorno et al., 2013) provided at higher resolution (about 1 km). The spatial resolutions indicated above refers to the resolution at which the data are provided but not to their effective resolutions. "

RC: p6, line 34: replace "extremity" by "end".

AA: Done

RC: p8, lines 4-5 should not be in italic.

AA: Done

RC: p8, line 10: "Modified": This is the first time this term appears. Could you explain it?

AA: This term correspond to the water mass named "Modified Atlantic Water (MAW)", this water mass name is no longer used. "Modified" has been deleted.

RC: p9, line 16: Remove here the comment about density: "During strong wind event, evaporation dominated and led to an upward (positive) water flux."

AA: Done

RC: p9, line 19: You may refer here to fig 2 (instead of fig 10a)?

AA: Done

RC: p10: See my main comment + Please, detail g,

AA: Done

 $\mathsf{RC: rho}_0, QnetandFw; Some sentences should not be initialic.$

AA: Done
RC: p12, line 14: "during IOP16b"

AA: Done

RC: p13, line 26: "the front is less marked" this is not so clear for me from figs 4 and 7. Are you talking about a smoothing? a reduction of the temperature difference?

AA: We are talking about a reduction of the density gradient. "the front is less marked" has been replaced by "the surface density gradient is less marked".

RC: p14, line 2: "For example, Thomas et al. (2016) show"

AA: Done

RC: p15: refs Drobinski et al. and Ducrocq et al. 2014; In addition, I think URLs in this "References" section must refer to the "doi.org" pages.

AA: The references are generated directly by Latex with the copernicus template.

RC: Figure 5: precise "observation" or "simulation" on each panel if possible. For (c), the caption tells it is the simulation bias but it seems to be more 'glider minus simulation'

AA: Corrected.

RC: Figure 7c: The red contour is not visible.

AA: The red contour is not drawn in the figure c. The caption has been modified.

RC: Figure 8 is difficult to read. Please consider here to plot separately the density sections for 25 oct., 30 oct. and the differences.

AA: Figure 8 has been modified for better visibility.

RC: Figure 10: Please, improve the resolution in order to distinguish the arrows. Precise in the caption if the arrows are for the wind or the wind stress + a reference length for vectors must be added.

AA: The resolution has been increased. The arrow corresponds to the wind stress, this
information added in the caption and a reference length for vectors is given.

RC: Figure 14: the plots of "B0+EBF" are not necessary

AA: The plots of "B0+EBF" has been deleted.